# LD-MoLE: Learnable Dynamic Routing for Mixture of LoRA Experts

**Yuan Zhuang**[1]*, **Yi Shen**[2]*, **Yuexin Bian**[3], **Qing Su**[1],
**Shihao Ji**[1], **Yuanyuan Shi**[3], **Fei Miao**[1]†

[1]University of Connecticut, [2]University of Pennsylvania, [3]University of California San Diego
```
{yuan.2.zhuang, qing.2.su, shihao.ji, fei.miao}@uconn.edu,
eshen6959@gmail.com,
{yubian, yyshi}@ucsd.edu
```

## Abstract

Recent studies have shown that combining parameter-efficient fine-tuning (PEFT) with mixture-of-experts (MoE) is an effective strategy for adapting large language models (LLMs) to the downstream tasks. However, most existing approaches rely on conventional TopK routing, which requires careful hyperparameter tuning and assigns a fixed number of experts to each token. In this work, we propose LD-MoLE, a **L**earnable **D**ynamic routing mechanism for Mixture of LoRA Experts that enables adaptive, token-dependent, and layer-wise expert allocation. Our method replaces the non-differentiable TopK selection with a differentiable routing function and a closed-form solution. Moreover, our design allows the model to adaptively determine the number of experts to activate for each token at different layers. In addition, we introduce an analytical sparsity control objective to regularize the number of activated experts. Extensive experiments on the Qwen3-1.7B and Llama-3.2-3B models show that LD-MoLE achieves the highest average scores compared to state-of-the-art baselines, across a diverse set of benchmarks. Our method not only achieves superior performance, but also demonstrates the ability to learn token-dependent and layer-wise expert allocation. The code is available at: https://github.com/eshentw/LD-MoLE.

## 1 Introduction

Large language models (LLMs) have demonstrated impressive capabilities across a wide range of natural language processing (NLP) tasks. However, their growing size requires significant computational resources for full-parameter fine-tuning. To address this, Parameter-Efficient Fine-tuning (PEFT) methods, such as Adapter-tuning (Houlsby et al., 2019) and LoRA (Hu et al., 2021), have emerged as crucial techniques for reducing training costs.

Recently, the Mixture-of-Experts (MoE) design (Jacobs et al., 1991; Shazeer et al., 2017) has been successfully integrated into transformer feed-forward networks during LLMs pretraining (Dai et al., 2024; Yang et al., 2025), demonstrating that MoE can reduce computational cost while maintaining strong performance. This has inspired a promising direction for PEFT, leading to the Mixture of LoRA Experts (MoLE) framework (Wu et al., 2024; Dou et al., 2024; Zadouri et al., 2023). MoLE utilizes multiple LoRAs as experts, providing a scalable and efficient alternative to relying on a single LoRA – where high-rank configurations risk overfitting and increased compute cost (Zhang et al., 2023), while low-rank ones often underperform (Liao et al., 2025; Gao et al., 2024).

Despite substantial advances, many recent MoE variants remain constrained by rigid routing strategies. A prominent example is MoLA (Gao et al., 2024), which relies on conventional TopK routing. This approach forces every token to consult a fixed number of experts, introducing a manually tuned hyperparameter that prevents adaptive allocation of resources based on token complexity. In

---

*These authors contributed equally to this work.

†Corresponding author: Fei Miao

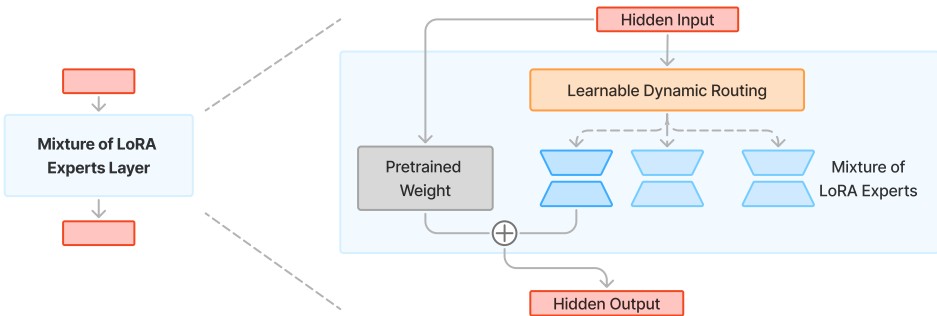

Figure 1: The overview of the LD-MoLE architecture, which enables Learnable Dynamic Routing (details in Section 3 and Fig 2 (c)) for LoRA adapters with the Mixture-of-Experts paradigm.

addition, the discrete and non-differentiable nature of the TopK operator hinders end-to-end optimization, ultimately limiting both performance and scalability (Shazeer et al., 2017; Zoph et al., 2022; Wang et al., 2025). Recent efforts such as ReMoE (Wang et al., 2025) attempt to bypass this bottleneck by replacing TopK with a ReLU-based router, but this dynamic scheme can suffer from instability, as some tokens may be routed to no experts at all, degrading overall performance. Taken together, these limitations highlight a central challenge: *Can we design a routing mechanism that adaptively learns to allocate experts in a stable and differentiable way?*

In this work, we propose LD-MoLE (see Figure 1), a **L**earnable and **D**ynamic routing method to adaptively control LoRA experts allocation. We adopt Sparsegen (Laha et al., 2018) as the projection onto the probability simplex and propose a dynamic routing mechanism and the corresponding training pipeline that has the following benefits: (1) the closed-form formulation with Sparsegen to decide routing probability ensures differentiability and guarantees that every token is assigned to at least one expert; (2) the routing admits a well-defined subgradient; (3) the derivative of the routing is upper-bounded, facilitating stable optimization; and (4) the routing design supports sparse yet controllable allocations. Building on this foundation, we introduce a lightweight, shared multi-layer perceptron (MLP) that predicts a token-specific sparsity parameter $\lambda$, governing expert selection. In addition, we formulate a sparsity control objective derived from Sparsegen's analytical solution, enabling direct regularization over the number of activated experts.

We conduct extensive experiments to validate the effectiveness of LD-MoLE. Specifically, we adopt Llama-3.2-3B and Qwen3-1.7B as base LLMs and fine-tune them on a wide range of instruction-tuning and sequence classification benchmarks. LD-MoLE achieves the best performance across these benchmarks, outperforming prominent baselines that follow different routing strategies – MoLA (Gao et al., 2024) with conventional TopK routing and ReMoE (Wang et al., 2025) with ReLU-based dynamic routing.These results indicate that our learnable routing mechanism yields consistent improvements across tasks and architectures. Moreover, we show that our sparsity control loss effectively reduces the number of activated experts without compromising performance.

Our contributions are threefold:

1. We propose LD-MoLE, a novel MoLE framework with an end-to-end learnable dynamic routing mechanism that adaptively allocates experts to tokens across layers.

2. We introduce an analytical sparsity loss, derived from the closed-form solution of Sparsegen, to explicitly regulate the number of activated experts.

3. We conduct comprehensive experiments on Llama-3.2-3B, Llama-3.2-8b and Qwen3-1.7B, including ablation studies and detailed analyses, to demonstrate the effectiveness of LD-MoLE and to elucidate the mechanisms behind its improvements over TopK and ReLU-based routing.

## 2 RELATED WORK

**Mixture of Experts.** MoE was first introduced in the 1990s (Jacobs et al., 1991) and later applied to large-scale neural networks (Shazeer et al., 2017) to efficiently scale up model capacity. Landmark models like Google's GShard (Lepikhin et al., 2020) implement sparse MoE frameworks with

Top2 expert routing, while Switch Transformer (Fedus et al., 2022) simplifies this to a single expert per token to reduce overhead. Today, MoE has been widely adopted in several well-known large language models, including GLaM (Du et al., 2022), Mixtral-8x7B (Jiang et al., 2024), DeepSeek-MoE (Dai et al., 2024), Qwen3 (Yang et al., 2025) and LongCat-Flash (Team et al., 2025).

**Routing Approaches in MoE.** Various routing strategies have been proposed for expert selection. The most common is TopK routing (Shazeer et al., 2017), where each token selects a fixed number of experts. There are also several works that discuss variants of TopK. AdaMoE (Zeng et al., 2024) uses conventional TopK routing with k larger than in vanilla MoE but achieves token-adaptive expert selection by incorporating null experts, which are defined as an empty operation. Ada-K Routing (Yue et al., 2024) introduces an allocator and then obtains $k^*$ for customized expert resource allocation instead of fixed TopK through a non-differentiable sampling operation with a RL-based optimization framework. Alternative designs, such as expert-choice routing (Zhou et al., 2022) reverse this perspective by allowing experts to select tokens. Beyond fixed-$k$ approaches, several methods aim to enable dynamic routing. For instance, TopP routing (Huang et al., 2024) selects experts until a cumulative probability threshold is reached, while DYNMOE (Guo et al., 2025) introduces Top-Any Gating to eliminate the need for tuning $k$. Soft MoE (Puigcerver et al., 2024) merges tokens and assigns them to experts as linear combinations, and Lory (Zhong et al., 2024) proposes a fully differentiable routing mechanism but underperforms TopK routing. Closest to our work, ReMoE (Wang et al., 2025) employs ReLU-based routing for differentiable and dynamic selection.

**Mixture of LoRA Experts.** Combining multiple LoRA modules (Hu et al., 2021) with MoE structure has led to the Mixture of LoRA Experts framework (Wu et al., 2024). Several variants have since been proposed: LoRAMoE (Dou et al., 2024) introduces MoE-style plugins to enhance downstream performance while mitigating knowledge forgetting; HMoRA (Liao et al., 2025) employs a hybrid scheme that hierarchically integrates token-level and task-level routing. MixLoRA (Li et al., 2024) builds a resource-efficient sparse MoE from LoRA modules. Other works, such as MoLA (Gao et al., 2024) and AlphaLoRA (Qing et al., 2024), analyze expert allocation patterns across layers. In this work, we introduce LD-MoLE, which integrates the fully differentiable Sparsegen formulation with a learned MLP to predict $\lambda$, enabling end-to-end dynamic expert routing.

## 3 APPROACH

As illustrated in Figure 2, we introduce a learnable dynamic routing mechanism that adaptively selects experts. Traditional MoE models (a) employ TopK routing, where each token is assigned to a fixed number of experts according to its top softmax scores. In contrast, our method (b) employs a closed-form routing formulation involving a token-dependent sparsity factor $\lambda$, predicted by a lightweight shared MLP, that controls the projection function, and thereby regulates the number of activated experts. This design enables the model to allocate more experts to tokens that demand greater modeling capacity and fewer to those that are easier to represent, effectively balancing efficiency and expressivity.

### 3.1 THE LD-MoLE ARCHITECTURE

In this section, we first review the TopK routing, then present our proposed Sparsegen routing with dynamic expert allocation, and finally describe how it is combined with LoRA to form the complete LD-MoLE architecture.

**TopK Routing.** The TopK router in a MoE layer determines the assignment of each token to the most suitable $k$ experts. In general, TopK routing computes a softmax distribution over the experts and calculates a weighted sum of the largest $k$ experts.

Formally, let $E$ and $d$ be the number of experts and input dimension respectively. We define the gate score weight matrix $\boldsymbol{W}_{\text{gate}} \in \mathbb{R}^{d \times E}$ and logits $\mathbf{u} \in \mathbb{R}^E$. The conventional TopK routing method takes token embedding $\mathbf{x}$ as inputs to predict the scores assigned to each expert:

$$\mathbf{u} = \boldsymbol{W}_{\text{gate}}\boldsymbol{x} \in \mathbb{R}^E, \tag{1}$$

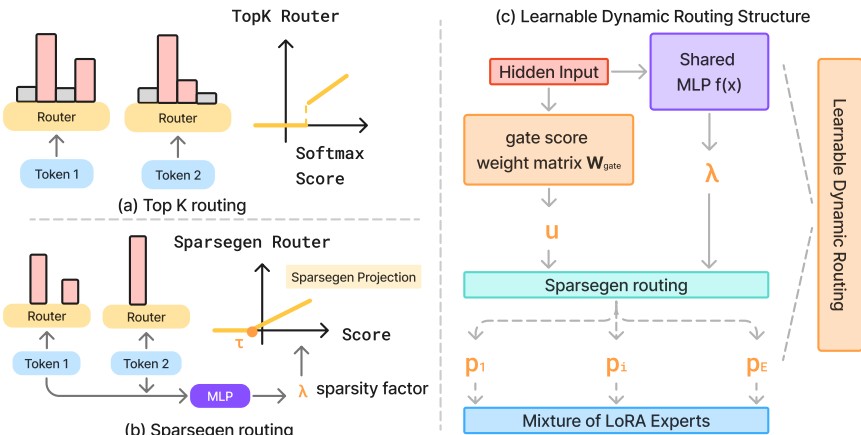

Figure 2: (a) Standard TopK routing activates a fixed number ($K$) of experts using non-differentiable selection. (b) Sparsegen routing introduces a differentiable projection onto the probability simplex, controlled by a sparsity parameter $\lambda$, which enables adaptive expert selection. (c) In the Sparsegen routing module, for each token, a lightweight shared MLP predicts the sparsity factor $\lambda$. Together with the logits $\mathbf{u}$, $\lambda$ determines the probability simplex $\mathbf{p}$ over LoRA experts, enabling dynamic, token-dependent expert allocation across layers. The detailed mathematical formulation is provided in Section 3.1.

Define $\mathcal{S}_k(\mathbf{u})$ as the index set of the TopK largest entries of $\mathbf{u}$. The routing output $\boldsymbol{p} \in \mathbb{R}^E$ is then given by

$$\boldsymbol{p}_i = \begin{cases} \dfrac{\exp(\boldsymbol{u}_i)}{\sum_{j \in \mathcal{S}_k(\mathbf{u})} \exp(\boldsymbol{u}_j)}, & i \in \mathcal{S}_k(\mathbf{u}), \\ 0, & \text{otherwise}, \end{cases}$$

which yields a sparse probability vector with at most $k$ nonzero entries. Note that the selection operator $\mathcal{S}_k(\cdot)$ introduces a jump discontinuity at the $k$-th largest value. Consequently, an arbitrarily small perturbation of the router scores can change the selected set and induce an abrupt change in the gradient, rendering the routing function non-differentiable at these boundaries.

Despite the success of conventional TopK routing with softmax operation in improving training and inference efficiency, two limitations persist (Guo et al., 2025; Wang et al., 2025): (1) TopK routing is non-differentiable during the learning process. (2) The value of $k$ requires carefully tuned to optimize model performance and would be fixed throughout the training process (Guo et al., 2025).

In contrast, Sparsegen (Laha et al., 2018) produces sparse routing weights via a closed-form projection, avoiding discrete TopK selection and yielding well-defined gradients that better align optimization with the routing behavior.

**Learnable Dynamic Routing.** To address the aforementioned limitations, we propose a *learnable dynamic* routing mechanism based on Sparsegen (Laha et al., 2018), which is a projection onto the probability simplex that generates sparse outputs via a closed-form and fully differentiable solution. Given the score vector $\boldsymbol{u}$ in Eq. 1, the routing function adaptively determines the effective number of activated experts by solving a closed-form transformation where $\lambda$ is a sparsity scalar:

$$\boldsymbol{p} = \operatorname*{argmin}_{\boldsymbol{p} \in \mathbb{R}^E} \|\boldsymbol{p} - \boldsymbol{u}\|_2^2 - \lambda \|\boldsymbol{p}\|_2^2, \qquad \text{s.t. } \boldsymbol{p} \geq 0, \ \mathbf{1}^\top \boldsymbol{p} = 1, \lambda < 1, \tag{2}$$

In our work, we introduce a lightweight MLP to predict a token-wise sparsity factor to control the degree of sparsity in the expert allocation, where $f$ denotes the shared MLP that produces a scaling coefficient $\lambda$ conditioned on $\boldsymbol{x}$:

$$\lambda = f(\boldsymbol{x}) \in \mathbb{R}, \tag{3}$$

The proposed routing function admits the following closed form.

**Proposition 1** (Closed-form Sparsegen routing: Proposition 0.1 in (Laha et al., 2018)). *Let* $\boldsymbol{u} \in \mathbb{R}^E$ *in Eq. 1 be the expert scores associated with token* $\boldsymbol{x}$, *and let* $\boldsymbol{u}_{(1)} \geq \cdots \geq \boldsymbol{u}_{(E)}$ *be the*

sorted coordinates of $\boldsymbol{u}$. Define the cumulative sums $U_k = \sum_{i=1}^{k} \boldsymbol{u}_{(i)}$ for $k = 1, \ldots, E$. Then the Sparsegen routing distribution $\boldsymbol{p} \in \mathbb{R}^E$ with sparsity parameter $\lambda \in (-\infty, 1)$ is given by

$$\boldsymbol{p}_i = \left[\frac{\boldsymbol{u}_i - \tau}{1 - \lambda}\right]_+, \quad \forall i \in [E], \tag{4}$$

where $[x]_+ = \max(x, 0)$, and the threshold $\tau$ is determined as

$$\tau = \frac{U_k - 1 + \lambda}{k}, \quad k = \max\{k \in [E] \,|\, 1 - \lambda + k\boldsymbol{u}_{(k)} > U_k\} \tag{5}$$

such that $\boldsymbol{p}$ lies on the probability simplex, i.e., $\sum_{i=1}^{E} \boldsymbol{p}_i = 1$.

*Proof.* This result follows from solving the Sparsegen projection problem, which minimizes a strongly convex objective subject to simplex constraints (Laha et al., 2018). □

As shown in Figure 2, the sparsity factor $\lambda$ and the input logits $\boldsymbol{u}$ jointly determine the threshold $\tau$, which defines the change point of the differentiable routing function. Intuitively, $\lambda$ controls the tendency toward sparsity in the solution. As $\lambda \to 1^-$, it pushes the distribution toward the simplex corners (sparse), while as $\lambda \to -\infty$, it drives the solution toward uniform simplex. Furthermore, we establish a key property of Sparsegen relevant to our setting:

**Lemma 1** (Sparsegen selects at least one expert.)**.** *Let $\boldsymbol{u} \in \mathbb{R}^E$ and $\lambda < 1$. The sparsegen solution (equation 2) always has nonempty support: $\|\boldsymbol{p}\|_0 \geq 1$.*

We provide a full proof for this lemma in Appendix A. Overall, the closed-form formulation in Proposition 1 offers both theoretical and practical advantages. It enables efficient computation of routing distributions and introduces a tunable sparsity factor $\lambda$, which allows the model to adaptively select a dynamic number of experts. Importantly, the routing remains fully differentiable, ensuring compatibility with end-to-end training.

**Model Layout.** In our work, we incorporate parameter-efficient LoRA adaptation into the MoE architecture. Each expert network is a LoRA module, where instead of updating the full weight matrix $\boldsymbol{W}_i \in \mathbb{R}^{d_{\text{out}} \times d_{\text{in}}}$, a low-rank update is introduced:

$$\Delta\boldsymbol{W}_i = \boldsymbol{A}_i\boldsymbol{B}_i, \quad \boldsymbol{A}_i \in \mathbb{R}^{d_{\text{out}} \times r}, \; \boldsymbol{B}_i \in \mathbb{R}^{r \times d_{\text{in}}}, \tag{6}$$

with $r \ll \min(d_{\text{out}}, d_{\text{in}})$. To adaptively capture non-linear relationships between token-level features, we employ a $\lambda_t$ (equation 3) predicted by a shared MLP in Fig 2 with $t = 1, \ldots, T$ for a sequence of $T$ tokens. The $\boldsymbol{x}_t$ is the token feature and would be the input for the shared MLP in Eq. 3. For each unique input size, we instantiate a single MLP, shared among all layers with that dimensionality. The shared MLP structure greatly reduces the number of additional parameters required while still allowing the router to predict $\lambda_t$ dynamically. This design decouples the parameter cost of predicting $\lambda_t$ from both the number of layers and the number of experts, leaving it dependent solely on the set of unique input dimensions. Given $\lambda_t$, the proposed router generates the routing weights $\boldsymbol{p}_t$ (equation 4) for each token and determines the weighted aggregation of the output embedding $\boldsymbol{h}_t$ from the LoRA-augmented experts:

$$\boldsymbol{h}_t = \boldsymbol{W}_{\text{base}}\boldsymbol{x}_t + \sum_{i=1}^{E} \boldsymbol{p}_{t,i}(\boldsymbol{A}_i\boldsymbol{B}_i\boldsymbol{x}_t). \tag{7}$$

Our framework also remains flexible: alternative MLP structures can be adopted, and we investigate a local variant in Appendix C.

## 3.2 TRAINING LOSS

In this work, we adopt the standard cross-entropy loss for the Language Model (LM) in both next-token prediction and sequence classification tasks (Xue et al., 2024; Liao et al., 2025; Wu et al., 2024; Dou et al., 2024). Formally, this could be expressed as

$$\mathcal{L}_{\text{LM}} = -\sum_{i=1}^{n+m} M_i \log P_{\text{LM}}(x_i \,|\, x_{<i}), \tag{8}$$

In this formulation, $X = (x_1, \ldots, x_{n+m})$ denotes the concatenation of the input sequence and target sequence with length $n$ and $m$ respectively. $M_i \in \{0, 1\}$ is a binary mask that specifies whether the $i$-th token contributes to the loss. In particular, $M_i = 0$ for tokens belonging to the input sequence(ignored during optimization), and $M_i = 1$ for tokens in the target sequence. This ensures that the model is trained to predict only the target tokens conditioned on both the input prompt and the previously generated target tokens, while not penalizing predictions over the input context.

To further stabilize the training process, we incorporate the conventional load-balancing loss (Fedus et al., 2022; Yang et al., 2025; Dai et al., 2024), which mitigates the risk of routing collapse (Shazeer et al., 2017). Such collapse can also arise in LoRA-augmented expert settings during fine-tuning, where only a few experts dominate the token assignments. Additionally, we introduce a sparsity loss that leverages the closed-form nature of our routing to directly regulate the sparsity level. In the following, we present the mathematical formulation of both the load-balancing loss and the proposed sparsity loss in detail.

### 3.2.1 LOAD BALANCING LOSS

Given $E$ experts indexed by $i = 1$ to $E$ and a batch $\mathcal{B}$ with $T = n + m$ tokens, the auxiliary loss is computed as the scaled dot-product between vectors $\mathcal{P}$ and $\mathcal{P}$,

$$\mathcal{L}_{\text{lb}} = E \cdot \sum_{i=1}^{E} \mathcal{F}_i \cdot \mathcal{P}_i \tag{9}$$

where $\mathcal{F}_i$ is the fraction of tokens dispatched to expert $i$, and $\mathcal{P}_i$ is the fraction of the router probability allocated for expert $i$,

$$\mathcal{F}_i = \frac{1}{T} \sum_{x \in \mathcal{B}} \mathbf{1}\{\text{Token } t \text{ selects Expert } i\}, \quad \mathcal{P}_i = \frac{1}{T} \sum_{x \in \mathcal{B}} \boldsymbol{p}_i(x). \tag{10}$$

This objective encourages both $\mathcal{F} = (\mathcal{F}_1, \ldots, \mathcal{F}_E)$ and $P = (P_1, \ldots, P_E)$ to approach a uniform distribution. In the ideal case of perfect balance, each expert receives an equal share, i.e., $\mathcal{F}_i = P_i = 1/E$ for all $i$, which minimizes Eq. 9. By penalizing concentration of both token assignments ($\mathcal{F}_i$) and router probabilities ($P_i$) on a small subset of experts, this simple yet effective loss plays a crucial role in ensuring stable and efficient MoE training.

### 3.2.2 CONTROLLING SPARSITY WITH SPARSITY LOSS

The proposed routing mechanism enables explicit control over sparsity via the predicted factor $\lambda$. To achieve a desired number of activated experts, we introduce a sparsity loss that regularizes $\lambda$ toward values corresponding to the target sparsity level.

Suppose we aim for exactly $k$ experts to be activated for a given token. From Proposition 1, this requires that the $k$-th largest score satisfies $\boldsymbol{u}_{(k)} > \tau$ while the $(k + 1)$-th largest score satisfies $\boldsymbol{u}_{(k+1)} \leq \tau$. This condition uniquely determines the target value range of $\lambda$ that yields $k$ activated experts. We formalize this in Proposition 2, which gives an analytical range of $\lambda$ that yields exactly $k$ activated experts.

**Proposition 2** (*k expert activation*). *Let $f(\boldsymbol{x}) = \boldsymbol{u} \in \mathbb{R}^E$ and let $\boldsymbol{u}_{(1)} \geq \cdots \geq \boldsymbol{u}_{(E)}$ be the sorted coordinates of $\boldsymbol{u}$, with $U_k$ defined as in Proposition 1. Then exactly $k$ experts are activated, i.e.,*

$$\boldsymbol{p}_{(i)} > 0, i \leq k, \quad and \quad \boldsymbol{p}_{(i)} = 0, i > k, \tag{11}$$

*if and only if the sparsity factor $\lambda$ lies in the interval*

$$\lambda \in \left[ 1 - \left( U_k - k\,\boldsymbol{u}_{(k+1)} \right), \, 1 - \left( U_k - k\,\boldsymbol{u}_{(k)} \right) \right), \quad 1 \leq k \leq E - 1. \tag{12}$$

*For $k = E$, the condition reduces to*

$$\lambda \in \left( -\infty, \, 1 - \left( U_E - E\,\boldsymbol{u}_{(E)} \right) \right). \tag{13}$$

*Proof.* The result follows by characterizing the threshold $\tau$ in Proposition 1 and enforcing the conditions $\boldsymbol{u}_{(k)} > \tau \geq \boldsymbol{u}_{(k+1)}$. We provide the detailed derivation in Appendix B. □

From Proposition 2, we define $\lambda_{\text{lower}}(k)$ as the lower bound of the interval in Eq. 12. When the goal is to maintain the number of selected experts less than or equal to $k$, motivate $\lambda$ to remain within this interval (with only lower bound) during training by introducing a sparsity loss of the form:

$$\mathcal{L}_{\text{sparse}} = \text{ReLU}(\lambda_{\text{lower}}(k) - \lambda) \qquad (14)$$

This loss penalizes $\lambda$ whenever it falls below the lower bound, while leaving it unchanged when $\lambda$ lies inside the feasible region.

Finally, altogether we optimize the following total loss objective with two coefficients $\alpha$ and $\beta$ that are hyperparameters to control the relative contribution of auxiliary losses:

$$\mathcal{L}_{\text{total}} = \mathcal{L}_{\text{LM}} + \alpha \, \mathcal{L}_{\text{lb}} + \beta \, \mathcal{L}_{\text{Sparse}}. \qquad (15)$$

## 4 EXPERIMENTS

### 4.1 EXPERIMENT SETUP

We evaluate our method by incorporating the MoE structure with Sparsegen routing in the Mixture of LoRA Experts setting to finetune the base model on various common benchmarks.

**Benchmarks and Metrics:** We evaluate the overall accuracy of our method against several baselines across a range of downstream tasks. Specifically, we test on standard NLP benchmarks, including instruction-finetuning datasets such as **MMLU-Pro** (Wang et al., 2024), **ARC-Challenge**, **ARC-Easy** (Clark et al., 2018), **OpenBookQA** (Mihaylov et al., 2018), **CommonsenseQA** (Talmor et al., 2019), **SWAG** (Zellers et al., 2018), **HellaSWAG** (Zellers et al., 2019), as well as sequence classification tasks from GLUE: **CoLA**, and **RTE** (Wang et al., 2019). For all benchmarks, we use standard accuracy as the evaluation metric. Please refer to Appendix E for more detail of the dataset and setup.

**Base Model and Baselines:** We test baseline approaches on different open-source LLMs, including Llama-3.2-3B, Llama-3.2-8b (Dubey et al., 2024) and Qwen3-1.7B (Yang et al., 2025). We compare our method primarily against MoLA (Gao et al., 2024), a TopK routing strategy within the MoLE framework, denoted as MoLA(8888). We also evaluate its proposed variant MoLA(2468), which assigns fewer experts to lower layers and progressively increases the allocation toward higher layers, reportedly yielding consistently better performance. In addition, we include ReMoLE, which adapts the ReLU-based routing from ReMoE (Wang et al., 2025) to the LoRA experts setting. Simliar to L2D-MoLE, ReMoLE supports both dynamic and differentiable routing.

**Implementation:** For our method, training is conducted on 4 NVIDIA H200 GPUs with a batch size of 16 for 10 epochs, with the learning rate of 0.0001 decayed by a factor of 0.1 at epochs 6 and 8. For Llama-3.2-8b, we run the experiment for only run 3 epochs as we observe a convergence at the early stage of training. We set the number of LoRA experts to 8, with rank 8 and scaling factor 16, and apply a dropout rate of 0.1. Across all methods, we pair the training with the load-balancing loss and follow the settings described in the original baseline papers. For MoLA, we choose the top 2 expert selections follows the original settings (Gao et al., 2024). For ReMoLE, we employ the load-balancing objective function introduced in ReMoE(Wang et al., 2025) with exact the same coefficients. More details of the method and experiment training setting are provided in Appendix D.

### 4.2 OVERALL PERFORMANCE

The overall performance of our proposed LD-MoLE is summarized in Table 1. Across all tested configurations, LD-MoLE achieves the highest average scores on both the Llama-3.2-3B, Llama-3.2-8b and Qwen3-1.7B models, demonstrating the consistent benefits of its learned dynamic routing. For this comparison, we set $\alpha = 1.0$ and disable the sparsity loss (i.e., $\beta = 0$), as defined in Eq. 15. A detailed analysis of sparsity control is deferred to Section 4.4.

In particular, LD-MoLE outperforms both fixed and dynamic routing baselines. Compared to the fixed TopK routing of MoLA, our method excels on reasoning-heavy benchmarks, achieving average cross-model gains of over +3.5% on ARC-E, SWAG, and HellaSWAG. For MMLU-Pro, LD-MoLE outperform the baselines in all testing models. On OpenBookQA, it achieves an average improvement of about +1.2%, and on CommonsenseQA, it surpasses MoLA by more than +2.0%. While

| Method | Model | TP | MMLU_P | ARC-C | ARC-E | Open | Comm | SWAG | Hella | CoLA | RTE | Avg |
|---|---|---|---|---|---|---|---|---|---|---|---|---|
| MoLA(8888) | Llama3.1-8B | 2.17 % | 48.21 | 76.79 | 86.32 | 86.40 | **84.11** | 89.15 | 94.06 | **87.73** | 89.53 | 82.48 |
| MoLA(2468) | Llama3.1-8B | 1.26 % | 50.37 | 77.90 | 87.37 | 87.60 | 83.87 | 86.97 | 94.03 | 87.25 | 90.25 | 82.85 |
| ReMoLE | Llama3.1-8B | 2.17 % | 54.98 | 82.00 | **91.61** | 87.60 | 83.17 | 92.22 | 94.95 | 85.27 | 88.43 | 84.47 |
| LD-MoLE | Llama3.1-8B | 2.33 % | **55.98** | **83.67** | **91.61** | **88.00** | 83.01 | **92.29** | **95.45** | 85.25 | **91.32** | **85.18** |
| MoLA(8888) | Llama3.2-3B | 3.11 % | 40.31 | 71.57 | 83.51 | 81.00 | 79.77 | 83.56 | 87.47 | 85.81 | **90.61** | 78.18 |
| MoLA(2468) | Llama3.2-3B | 1.80 % | 42.31 | 71.91 | 83.86 | 83.60 | 80.02 | 83.96 | 87.31 | 86.00 | 89.53 | 78.72 |
| ReMoLE | Llama3.2-3B | 3.11 % | 48.01 | **75.25** | 89.30 | 83.40 | 79.52 | 90.45 | 93.44 | 83.95 | 89.46 | 81.42 |
| LD-MoLE | Llama3.2-3B | 3.28 % | **49.58** | 74.58 | **89.47** | **84.00** | **81.42** | **91.37** | **93.60** | **86.02** | 88.38 | **82.05** |
| MoLA(8888) | Qwen3-1.7B | 4.12 % | 51.12 | 76.59 | 88.60 | 82.40 | 76.49 | 84.11 | 83.35 | **83.89** | 86.64 | 79.24 |
| MoLA(2468) | Qwen3-1.7B | 2.39 % | 49.96 | 76.92 | 88.42 | 83.00 | 75.84 | 84.17 | 87.09 | 83.60 | 84.48 | 79.28 |
| ReMoLE | Qwen3-1.7B | 4.12 % | 53.40 | **79.60** | 91.75 | 84.80 | **79.44** | 86.37 | 88.00 | 82.12 | 83.74 | 81.02 |
| LD-MoLE | Qwen3-1.7B | 4.23 % | **53.82** | 78.67 | **92.11** | **85.00** | 79.30 | **86.72** | **88.71** | 82.61 | **87.72** | **81.63** |

Table 1: Comparison between methods across downstream tasks.

MoLA attains slightly better results on certain sequence classification tasks such as RTE, LD-MoLE consistently delivers higher overall averages, underscoring the effectiveness of learnable dynamic routing across diverse task types. Compared to ReMoLE, LD-MoLE achieves higher overall averages, including +0.5% on Llama3-2.3B and +0.6% on Qwen3-1.7B. Notably, ReMoLE exhibits large performance drops on CoLA with Llama3-2.3B and RTE with Qwen3-1.7B, whereas LD-MoLE maintains stable effectiveness across benchmarks.

We observe that dynamic routing methods generally perform better on instruction fine-tuning tasks, while fixed routing approaches show slight advantages in certain sequence classification tasks. A possible explanation is that classification tasks often benefit from more uniform expert usage, where fixed routing ensures stable allocation. Interestingly, the pruned variant MoLA(2468) outperforms the standard MoLA(8888), suggesting that many experts in the fixed routing setup are underutilized, introducing redundancy as also noted in their work (Gao et al., 2024). In contrast, dynamic routing adapts expert selection on a token-by-token basis, which benefits complex reasoning and instruction-following tasks but may introduce variability that is less advantageous for shorter classification settings. Overall, LD-MoLE provides a stronger balance between parameter efficiency and performance, adapting better effectiveness across diverse tasks.

### 4.3 PREDICTED $\lambda$ FOR DYNAMIC EXPERT ALLOCATION

In this section, we compare the performance of the predicted $\lambda$ against fixed $\lambda$ values to demonstrate that the shared learnable MLP structure proposed for the prediction $\lambda$ achieves superior results in the setting of LoRA experts. We conduct experiments with Qwen3-1.7B as the base model and report results in Table 2. We evaluate a range of fixed $\lambda$ values against our predicted ones. Recall our routing formulation (equation 2) and Proposition 1, the parameter $\lambda$ directly controls the sparsity of the probability distribution over LoRA experts.

We provide the visualization of $\lambda$ value distribution with 25–75 quantile range in Figure 3 for K projection, gate projection and down projection module. We observe that the distribution of $\lambda$ varies substantially between layers. The value increases in magnitude and exhibits greater variance at deeper layers and that fixed $\lambda$ cannot capture this depth-wise variance. Motivated by this observation, we tested our predicted $\lambda$ against fixed values ranging from $0.5$ to $-10.0$. The results show that the predicted $\lambda$ consistently outperforms all fixed settings, demonstrating its ability to dynamically adapts across both layers and tokens. Naturally, a predicted $\lambda$ could flexibly adjusts without per-task or per-layer hyperparameter tuning thus yields improved performance against fixed $\lambda$.

### 4.4 SPARSITY CONTROL ANALYSIS

In this section, we evaluate how the proposed sparsity loss influences the expert pattern and impacts task performance on 5 datasets. To encourage sparsity, we set the target number of activated experts to $\leq 2$. Results show that the sparsity loss effectively reduces the overall number of activated experts. Increasing the sparsity alignment coefficient enforces a stronger constraint on the admissible range of $\lambda$ in Eq. 12. Moreover, Figure 4 illustrates the effect of applying the sparsity loss. Normally, more experts are activated in the lower layers, with activations gradually decreasing toward higher layers. Stronger regularization further suppresses higher-layer activations, while lower layers re-

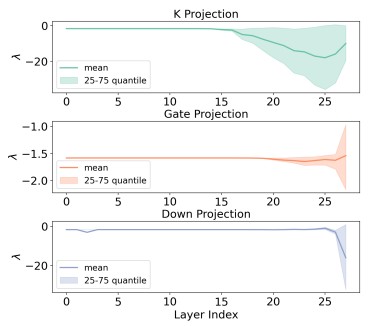

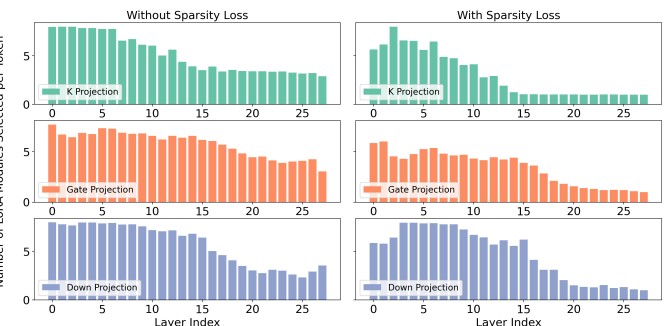

Figure 3: Layer-wise $\lambda$ values for K, gate, and down projections.

Figure 4: Average number of LoRA experts selected per token across layers.

| $\lambda$ | ARC-C | ARC-E | Open | Comm | RTE | Avg. |
|---|---|---|---|---|---|---|
| 0.5 | 77.92 | 91.93 | 82.60 | 78.13 | **87.72** | 83.66 |
| -1.0 | 77.26 | 91.93 | 83.80 | 78.38 | 86.17 | 83.50 |
| -10.0 | 77.26 | **92.11** | 83.40 | 78.71 | 85.29 | 83.35 |
| Predicted | **78.67** | **92.11** | **85.00** | **79.30** | **87.72** | **84.56** |

| Coeff | ARC-C | ARC-E | Open | Comm | RTE | Avg. |
|---|---|---|---|---|---|---|
| 1.0 | 76.25 | 90.88 | 82.60 | 77.15 | **88.69** | 83.11 |
| 0.1 | 76.92 | **92.28** | 82.80 | 79.03 | 87.30 | 83.66 |
| 0.01 | 78.26 | 91.40 | 84.20 | 78.79 | 87.47 | 84.02 |
| 0.0 | **78.67** | 92.11 | **85.00** | **79.30** | 87.72 | **84.56** |

Table 2: Quantative result on different $\lambda$ values for sparsity loss (Qwen3-1.7B).

Table 3: Quantitative results on different coefficient values for sparsity loss (Qwen3-1.7B).

main relatively dense. As shown in Table 3, the results emphasize a clear trade-off between sparsity and task performance: disabling the sparsity loss achieves the best average score, yet certain tasks benefit from reduced expert usage, suggesting that the optimal sparsity level is task-dependent. Our main contribution in this aspect is not simply confirming the sparsity and performance trade-off, but demonstrating that LD-MoLE makes sparsity both controllable and learnable within a dynamic routing framework. This enables the number of activated experts to be reduced in a principled way, while maintaining competitive performance. We also provide an additional experiment of computational analysis for this loss with respect to different hyperparameter $\beta$ in Appendix C.3.

At the same time, excessive sparsity can degrade performance, as we observe performance drops in the later stages of training. This suggests that enforcing too much sparsity beyond a certain point restricts flexibility in expert usage across layers, ultimately limiting performance. Overall, maintaining a balance dynamic system appears most effective for improving efficiency without undermining model capability.

### 4.5 Harder token need more experts

Dynamic routing allows the model to flexibly dedicate more capacity to tokens that require richer representations while conserving resources on more frequent or predictable ones. As shown in Figure 5, tokens that frequently appear during training (e.g., prompt- and context-related tokens) tend to activate fewer experts, effectively compressing their representations. In contrast, rarer or less familiar tokens activate a larger and more diverse set of experts, suggesting that tokens requiring greater modeling capacity benefit from richer expert combinations. This behavior is consistent with observations reported in ReMoE (Wang et al., 2025) for MoE pretraining. Overall, such adaptive routing enables the model to balance computation efficiently across tokens, allocating more resources to rare or informative ones.

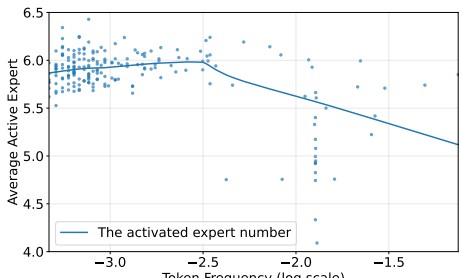

Figure 5: Correlation between the frequency of the top 200 most common tokens and their average number of activated experts. Each scatter point represents the average number of experts activated for a given token.

## 5 MORE ABLATION STUDIES

**Ablation study on zero-activation problem:** In Appendix C.1, we provide a detailed analysis of the zero-activation issue in dynamic routing. In particular, we show that ReMoLE can assign zero experts to a token, which leads to degenerate representations, whereas our method guarantees at least one expert is activated through the closed-form Sparsegen routing.

**Ablation study on expert patterns during training:** In Appendix C.2, we show that expert activation patterns are largely established early in training and remain fixed thereafter.

**Additional exploration of LD-MoLE:** In Appendix C.3, we present further experiments and discussions on our method. We provide an alternative local MLP design and analyze the impact of varying hidden dimensions in the shared MLP. In addition, we present experimental results on efficiency.

## 6 CONCLUSION

In this work, we introduce LD-MoLE, a learnable dynamic routing method for Mixture of LoRA Experts. Building on Sparsegen, our approach leverages a shared MLP to learn the sparsity parameter $\lambda$, enabling adaptive expert allocation across layers and tokens in a parameter-efficient manner. Comprehensive experiments show that LD-MoLE achieves the highest average scores on both the Llama-3.2-3B, Llama-3.2-8b and Qwen3- 1.7B models compared to strong baselines, including TopK routing and ReLU-based routing, across a range of instruction-tuning and sequence classification tasks. For future research, we want to see how LD-MoLE performs in the pretraining stages of LLMs with its differentiability and controllable sparsity. Furthermore, integrating our dynamic routing framework with other PEFT methods or extending its applicability to new domains, such as multi-modal models, presents exciting opportunities for future exploration.

## 7 ACKNOWLEDGEMENTS

The work of Yuan Zhuang and Fei Miao is partly supported by the National Science Foundation under Grants CNS-2047354 (CAREER). We also gratefully acknowledge the generous support of a Qualcomm gift fund and our collaborator Jonathan Petit.

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

# A    PROOF FOR LEMMA 1

*Proof.* When $\lambda < 1$, the closed form of sparsegen is

$$p_i = \left[ \frac{u_i - \tau}{1 - \lambda} \right]_+, \qquad i = 1, \ldots, E,$$

where $\tau$ is chosen so that $\sum_{i=1}^{E} p_i = 1$. Since each term is nonnegative and their sum equals 1, at least one term must be strictly positive. Hence the support $S(\boldsymbol{u}) = \{i : p_i > 0\}$ is nonempty and $\|\boldsymbol{p}\|_0 \geq 1$.

Equivalently, using the support-size characterization, let $u_{(1)} \geq u_{(2)} \geq \cdots \geq u_{(E)}$ be the sorted coordinates and $U_k = \sum_{i=1}^{k} \boldsymbol{u}_{(i)}$. From Eq. 5

$$k = \max \left\{ k \in [E] \mid 1 - \lambda + k\, u_{(k)} > U_k \right\}.$$

For $k = 1$ the inequality reduces to $1 - \lambda > 0$, which holds when $\lambda < 1$. Thus $k \geq 1$, so at least one index is selected.

For the edge case $\lambda = 1$, the quadratic term vanishes and the objective reduces to a linear program:

$$\max_{\boldsymbol{p} \in \mathbb{R}^E} \; \boldsymbol{p}^\top \boldsymbol{u}, \qquad \text{s.t. } \boldsymbol{p} \geq 0, \; \mathbf{1}^\top \boldsymbol{p} = 1.$$

Its maximizer is any one-hot vector supported on $\arg\max_i u_i$. Again, $\|\boldsymbol{p}\|_0 = 1$.

In all cases with $\lambda \leq 1$, the optimizer $\boldsymbol{p}$ is feasible ($\boldsymbol{p} \geq 0$, $\mathbf{1}^\top \boldsymbol{p} = 1$). A feasible vector on the simplex cannot be identically zero, hence its support is nonempty. $\qquad\square$

# B    PROOF FOR PROPOSITION 2

*Proof.* From Proposition 1, the routing probabilities are

$$\boldsymbol{p}_i = \left[ \frac{\boldsymbol{u}_{(i)} - \tau}{1 - \lambda} \right]_+,$$

where $\tau$ defined in (equation 5). For exactly $k$ experts to be activated, we require

$$\boldsymbol{u}_{(k)} > \tau \quad \text{and} \quad \boldsymbol{u}_{(k+1)} \leq \tau.$$

Substituting (equation 5), the first inequality gives

$$\boldsymbol{u}_{(k)} > \frac{U_k - 1 + \lambda}{k} \quad \Longleftrightarrow \quad \lambda < 1 - (U_k - k\boldsymbol{u}_{(k)}).$$

Similarly, the second inequality gives

$$\boldsymbol{u}_{(k+1)} \leq \frac{U_k - 1 + \lambda}{k} \quad \Longleftrightarrow \quad \lambda \geq 1 - (U_k - k\boldsymbol{u}_{(k+1)}).$$

Combining the two inequalities, we obtain

$$\lambda \in \left[ 1 - (U_k - k\boldsymbol{u}_{(k+1)}), \; 1 - (U_k - k\boldsymbol{u}_{(k)}) \right),$$

which establishes (equation 12) for $1 \leq k \leq E - 1$.

For the case $k = E$, only the condition $\boldsymbol{u}_{(E)} > \tau$ applies. Substituting again yields

$$\boldsymbol{u}_{(E)} > \frac{U_E - 1 + \lambda}{E} \quad \Longleftrightarrow \quad \lambda < 1 - (U_E - E\boldsymbol{u}_{(E)}).$$

$\qquad\square$

## C  ADDITIONAL EXPERIMENT RESULTS

### C.1  THE ZERO-ACTIVATION ISSUE IN DYNAMIC ROUTING

A key challenge in designing dynamic and differentiable routing mechanisms is the possibility of *zero activation*, where a token is not assigned to any expert. This problem occurs when the routing function produces zero outputs, leaving the token without an activated expert. Such cases not only waste model capacity but also hinder gradient flow, making it difficult for the affected experts to learn meaningful representations.

This issue arises in activation-based gating mechanisms such as ReLU-based routing, where the router may output all non-positive values for certain tokens. In practice, this leads to suboptimal expert utilization: some tokens receive no expert processing, while others may be redundantly assigned. Figure 6 compares the expert activation patterns of LD-MoLE and ReMoLE. From Figure 6, ReMoLE shows a similar trend to LD-MoLE, it activates more experts in the lower layers and fewer in the higher layers. However, its higher layers often fall below average 1.0 activated experts. This indicates that, for some tokens, the ReLU-based router fails to activate any experts in the upper layers.

In contrast, our proposed L2D-MoLE framework guarantees at least one routing coefficient remains strictly positive for every token. This ensures that all tokens are processed by at least one expert, while still enabling dynamic and sparse expert allocation across layers.

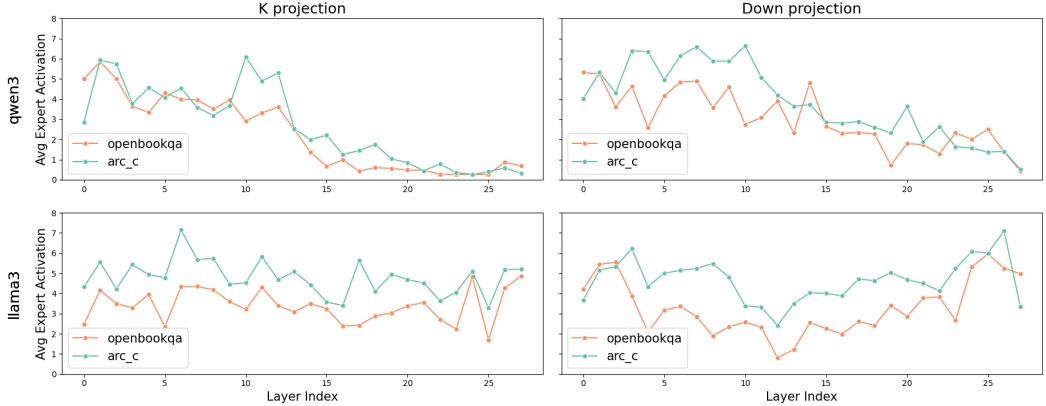

Figure 6: The average expert activation for ReMoLE on K and Down Projection modules. The green and orange line indicate the activation pattern on OpenbookQA and ARC-Chanllenge dataset respecitively.

### C.2  EXPERT PATTERN DURING TRAINING

In Fig. 7, we compare the expert activation patterns at the first and final training epochs. The trend described in Sec 4.4 which more experts are activated in the lower layers, with a gradual decrease toward the higher layers has already established by the end of the first epoch. The distribution remains largely consistent throughout training, as shown by the similarity between the heatmaps of routing ratio for Epoch 1 and Epoch 10. This indicates that routing specialization emerges very early and stabilizes quickly, leaving little room for substantial redistribution across layers as training progresses. Such stability underscores the importance of the early training phase: the model rapidly learns how to allocate experts, and subsequent optimization primarily fine-tunes within this established structure rather than reshaping it.

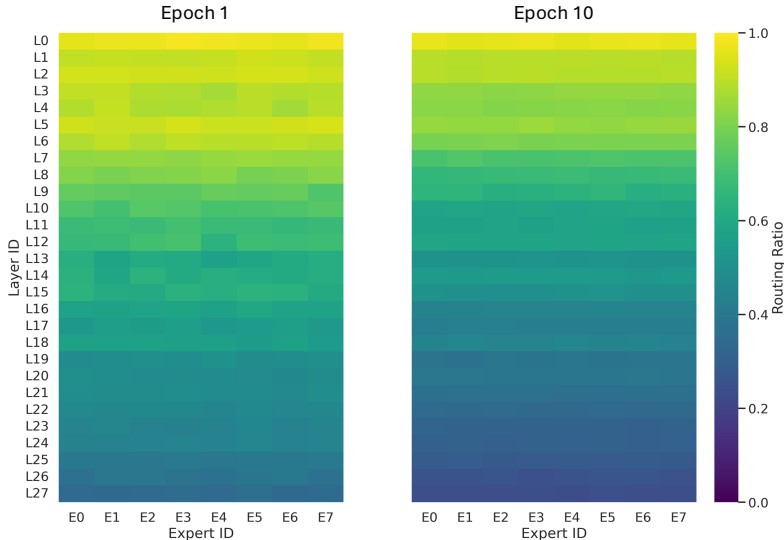

Figure 7: Comparision of the routing ratio heatmap of the expert activation pattern between the epoch 1 and epoch 10.

| Method | Model | TP | ARC-C | ARC-E | Open | Comm | SWAG | HellaSWAG | CoLA | RTE | Avg |
|---|---|---|---|---|---|---|---|---|---|---|---|
| Local | Llama3.2-3B | 3.13 % | 73.67 | **89.65** | 83.80 | **81.59** | 91.29 | 93.50 | 84.28 | **89.70** | 85.93 |
| Shared | Llama3.2-3B | 3.28 % | **74.58** | 89.47 | **84.00** | 81.42 | **91.37** | **93.60** | **86.02** | 88.38 | **86.10** |
| Local | Qwen3-1.7B | 4.14 % | 78.00 | 91.75 | 84.00 | **79.38** | **87.02** | 88.55 | **82.67** | 85.88 | 84.65 |
| Shared | Qwen3-1.7B | 4.23 % | **78.26** | **92.11** | **85.00** | 79.30 | 86.72 | **88.71** | 82.61 | **87.72** | **85.05** |

Table 4: Comparison between shared and local MLP structure for LD-MoLE.

## C.3 ADDITIONAL EXPLORATION ON LD-MOLE

**Shared vs Local MLP**
In Sec. 4, we presented results using the shared MLP design for learning the parameter $\lambda$. Here, we investigate an alternative architecture in which, instead of instantiating one expert MLP per unique input dimension (as described in Sec. 3.1), we assign a dedicated MLP to every target module (i.e., Q, K, V, O, Up, Gate, and Down projections). This design allows each module to learn its own specialized routing strategy, which is intuitively reasonable since different modules process distinct types of information. However, this approach significantly increases the number of tunable parameters, as modern LLMs contain hundreds of such modules. To mitigate this overhead, we restrict each local MLP to a single linear layer, such that $f$ in Eq. 3 reduces to a weight matrix $W_{\mathrm{mlp}} \in \mathbb{R}^{d_{\mathrm{in}} \times 1}$. But still, unlike the shared MLP design, the trainable parameters of the local MLP structure would be associated with the layer number of pretrained transformer models.

We report the comparison between shared and local structures on Qwen3-1.7B and Llama3-2.3B in Table 4. Results show that the local design adds fewer additional parameters than the shared counterpart, but overall achieves weaker performance. While the local MLP occasionally outperforms the shared version on certain datasets, the gains are marginal. This suggests that although local MLPs can individually learn $\lambda$, its limited capacity that using only a single linear transformation hinders their ability to fully capture the complexity of routing decisions.

**Hidden Dimension in Shared MLP**
In Sec. 4, we only use one MLP per unique input dimension. For example, Qwen3-1.7B contains seven modules but only two distinct input sizes (2048 and 6144), so only two MLPs are required. We set the hidden size of the MLPs to 256 for Qwen3-1.7B and 512 for Llama-3.2-3B. Here, we provide

| Dimension | Model | TP | ARC-C | ARC-E | Open | Comm | RTE | Avg |
|-----------|-------|-----|-------|-------|------|------|-----|-----|
| 128 | Llama3.2-3B | 3.15 % | 73.91 | 88.77 | 82.20 | 81.51 | **90.28** | 83.33 |
| 256 | Llama3.2-3B | 3.20 % | 72.24 | **89.65** | 82.80 | 81.18 | 88.23 | 82.82 |
| 512 | Llama3.2-3B | 3.28 % | **74.58** | 89.47 | **84.00** | **81.42** | 88.38 | **83.57** |
| 128 | Qwen3-1.7B | 4.17 % | 76.59 | 91.75 | 84.00 | 79.46 | 86.68 | 83.66 |
| 256 | Qwen3-1.7B | 4.23 % | **78.67** | **92.11** | **85.00** | 79.30 | 87.72 | **84.56** |
| 512 | Qwen3-1.7B | 4.34 % | 77.59 | 91.58 | 83.40 | **79.54** | **88.23** | 84.07 |

Table 5: Comparison between different hidden dimension (128, 256 and 512) used in LD-MoLE MLP.

comprehensive results across five datasets using various hidden dimensions in Table 5. The results show that performance peaks at 256 for Qwen3-1.7B and 512 for Llama3.2-3B, suggesting that each base model has an optimal hidden dimension. A plausible explanation is the difference in input size across models. For Llama3.2-3B, the module dimensions are larger, requiring a higher-capacity MLP (larger hidden dimension) to effectively capture the meaningful information and relationships needed for routing. Conversely, for Qwen3-1.7B, a smaller hidden dimension is sufficient, as overly large MLPs may introduce redundancy and lead to diminishing returns. Therefore, selecting the hidden dimension should balance representation capacity, parameter efficiency, and generalization ability.

**Additional Experiment for the Sparsity Loss:**

In this section, we compare the computational cost (in FLOPs) under different hyperparameter settings for our analytical loss function (Eq. 15). As shown in Table 6, the FLOP analysis further highlights the efficiency of the sparsity loss introduced in Section 3.2.2. As the sparsity-loss coefficient $\beta$ increases, the number of activated LoRA experts decreases, leading to a notable reduction in overall FLOPs. However, compared with conventional TopK and ReLU routing, the primary source of additional computation in our method arises from the shared MLP used to predict each token's sparsity factor $\lambda$. To further mitigate this overhead, a promising direction is to augment an additional dimension into the gating projection for generating $\lambda$ and we leave it to the future exploration.

| Qwen3-1.7B | MoLA-8888 | MoLA-2468 | ReMoLE | Ours($\beta = 1.0$) | Ours($\beta = 0.1$) | Ours($\beta = 0.01$) | Ours($\beta = 0$) |
|-----------|-----------|-----------|--------|---------------------|---------------------|----------------------|-------------------|
| MFLOPs | 43 | 40 | 74 | 83 | 90 | 100 | 106 |

Table 6: Effective FLOPs (router + LoRA experts) across different $\beta$ parameter for the sparsity loss and different routing baseline. Backbone FLOPs are excluded since they are identical across methods.

# D HYPERPARAMETER AND TRAINING SETUP

**Training Setup:** To ensure fairness, we adopt a consistent parameter-tuning pipeline and apply identical prompts across all datasets and methods. For instruction-tuning tasks, we mask out the prefix and context, training only on the final answer tokens. For sequence classification tasks, since LLMs lack a dedicated classification or separator token, we omit the former and replace the latter with the end-of-sentence token to mark sentence boundaries.

**Hyperparameters:** Table 7 summarizes the hyperparameter configurations used across different routing methods. To ensure fairness, we keep most training settings consistent, including optimizer (AdamW), batch size (16), and number of epochs (10). All methods are trained with LoRA rank $r = 8$, scaling factor $\alpha = 16$, and 8 experts. We also apply the method on all the target modules (i.e., Q, K, V, O, Up, Gate, and Down projections). For optimization, we adopt different learning rate schedules to align with prior works. Both LD and ReMoLE use the MultiStepLR scheduler with an initial learning rate of $1 \times 10^{-4}$, while MoLA follows its original implementation with cosine annealing and a slightly higher learning rate ($3 \times 10^{-4}$). This setup provides a balanced comparison by respecting the design choices of each baseline while maintaining comparable training stability. Dropout is applied to mitigate overfitting. LD-MoLE and ReMoLE use a dropout of 0.1,

Table 7: Hyperparameters used for different methods.

| Method | LD-MoLE | ReMoLE | MoLA |
|---|---|---|---|
| Cutoff Length | 1024 | 1024 | 1024 |
| Learning Rate | 1e-4 | 1e-4 | 3e-4 |
| scheduler | MultiStepLR | MultiStepLR | CosineAnneal |
| Optimizer | AdamW | AdamW | AdamW |
| Batch size | 16 | 16 | 16 |
| Dropout | 0.1 | 0.1 | 0.05 |
| Epochs | 10 | 10 | 10 |
| Target Modules | All | All | All |
| Routing type | Dynamic | Dynamic | Fixed |
| LoRA Rank $r$ | 8 | 8 | 8 |
| LoRA Alpha $\alpha$ | 16 | 16 | 16 |
| Experts | 8 | 8 | 8 |
| TopK | - | - | 2 |

whereas MoLA uses 0.05, again consistent with its reported configuration. The cutoff length for all experiments is fixed to 1024 to ensure uniform input context across models.

## E  DATASET INFORMATION

In this section, we provide additional details about the datasets and experimental setup. Each dataset is divided into three splits: training, validation, and test. Our experiments are conducted by training on the training split and evaluating on the validation split, without using the test split.

**ARC (AI2 Reasoning Challenge):**  ARC is a benchmark of grade-school level science questions with 4 choices, divided into two subsets: ARC-Easy, which consists of relatively straightforward questions, and ARC-Challenge, which requires more complex reasoning and deeper scientific knowledge. For ARC-Easy, there are 2251 samples in train split, 570 samples in validation split and 2376 samples in test splits. For ARC-Challenge, there are 1119 samples in train split, 299 samples in validation split and 1172 samples in test splits.

**CommonsenseQA:** CommonsenseQA is a multiple-choice question answering dataset with 5 choices that evaluates a model's ability to apply various forms of commonsense knowledge. It consists of 12,102 questions, each with one correct answer and four distractors. There are 9741 samples in train split, 1221 samples in validation split and 1140 samples in test splits.

**OpenBookQA:** OpenBookQA is designed to advance research in complex question answering with 4 choices by evaluating both scientific knowledge and language understanding. The dataset is modeled after open-book exams: it provides a collection of scientific facts that must be combined with broader commonsense knowledge to answer multiple-choice questions. Unlike simple fact-retrieval tasks, OpenBookQA emphasizes multi-step reasoning, integration of external knowledge, and deeper text comprehension. There are 4957 samples in train split, 500 samples in validation split and 500 samples in test splits.

**SWAG:** This benchmark evaluates commonsense reasoning by asking the model to predict the most plausible continuation of a given scenario. Each instance is formulated as a 4-way multiple-choice question, with one correct answer and three adversarially generated distractors. There are 73546 samples in train split, 20006 samples in validation split and 20005 samples in test splits.

**HellaSWAG:** It's designed to evaluate a model's ability to complete sentences in a coherent and contextually appropriate way. Similar to SWAG, each examples has 4 options or candidate endings, where the task is to select the most plausible continuation. The challenge lies in the fact that success requires more than recognizing surface-level word patterns—it demands an understanding of meaning, context, and commonsense reasoning. While this task is trivial for humans with extensive real-world and linguistic experience, it remains a significant hurdle for machines. There are 39900 samples in train split, 10000 samples in validation split and 10000 test samples.

**MMLU-Pro:** It is designed to evaluate a model's multi-domain knowledge understanding and complex reasoning abilities. Compared to MMLU, MMLU-Pro introduces more challenging, reasoning-focused questions and expands the answer choices from 4 to 10 options, while removing trivial and noisy items. Models using Chain-of-Thought reasoning perform better, reflecting the benchmark's emphasis on deeper reasoning. The dataset contains 12,102 samples.

**CoLA(Corpus of Linguistic Acceptability):** It's part of the General Language Understanding Evaluation(GLUE) benchmark and it consists of 10,657 sentences drawn from 23 linguistics publications, each annotated for grammatical acceptability by the original authors. The public release includes 9,594 sentences for training and development, while 1,063 test sentences are held out.

**RTE(Recognizing Textual Entailment):** It's part of the General Language Understanding Evaluation(GLUE) benchmark is consist of a series of annual entailment challenges. Examples are drawn from news and Wikipedia text. All datasets are converted into a two-class classification: entailment vs. not entailment, containing 2,490 training, 277 validation and 3000 test samples.

