# OpenReview forum: "LD-MoLE: Learnable Dynamic Routing for Mixture of LoRA Experts"
_ICLR.cc/2026/Conference — ICLR 2026 Poster_

### Official Review · Reviewer_oEkd · 2025-10-30

**Soundness:** 3
**Presentation:** 3
**Contribution:** 2
**Rating:** 4
**Confidence:** 4

**Summary:**

This paper introduces LD-MoLE, a Mixture-of-LoRA-Experts framework with learnable dynamic routing based on a closed-form Sparsegen function. A shared MLP predicts a token-wise sparsity factor λ , allowing the model to dynamically adjust the number of activated experts per token in a differentiable and stable manner. The method also includes an analytical sparsity loss for expert control.

**Strengths:**

1. Originality - The paper proposes a Sparsegen-based dynamic routing method with a mathematically proven sparsity loss, showing novelty and clear motivation.

2. Quality and Clarity – The methodology is well-designed and theoretically sound, with clear writing that makes the approach easy to follow.

3. Significance – The visualization of LoRA experts selected per token clearly demonstrates the effectiveness of the proposed dynamic routing mechanism.

**Weaknesses:**

1. Insufficient Ablation Studies and Experiments
The paper lacks an ablation study on the load balancing loss weight and its impact on the model's performance. It would be beneficial to explore how varying this parameter affects the stability and performance of the model, especially since load balancing plays a crucial role in expert distribution.
The paper also does not compare against important baselines like:
[1] Harder Task Needs More Experts: Dynamic Routing in MoE Models
[2] HMoRA: Making LLMs More Effective with Hierarchical Mixture of LoRA Experts Including these baselines would provide a more comprehensive evaluation and highlight the specific advantages of the proposed approach.

2. Lack of Large-Scale Model Comparison
The experiments are limited to relatively smaller models (Llama-3.2-3B and Qwen-3-1.7B). A comparison against larger-scale models would help validate the scalability of the proposed method and its ability to handle more complex tasks. This would provide a clearer picture of how the method performs in more resource-intensive settings.

3. Unconventional Benchmark Selection
The choice of experimental benchmarks seems somewhat unusual. The paper compares against a small subset of tasks, whereas other multi-task MoLE baselines (e.g., [3] KASA: Knowledge-Aware Singular-Value Adaptation of Large Language Models and [4] MTL-LoRA: Low-Rank Adaptation for Multi-Task Learning) involve a broader range of tasks. The rationale behind selecting this specific subset is unclear, and a more diverse set of benchmarks could provide a more robust evaluation. A clarification on why this subset was chosen would improve the overall experiment design.

4. Unclear Necessity of Sparsegen Routing
The relationship between Sparsegen routing and the Mixture-of-Experts (MoE) architecture is not fully explained. It is unclear why Sparsegen routing was specifically chosen over other potential methods. Has the paper explored alternatives like Gumbel-Softmax or directly using softmax without Top-K selection? These methods could provide valuable insights and potentially simplify the routing process. A discussion on why Sparsegen is preferred, and a comparison to other discrete or continuous routing methods, would strengthen the theoretical justification for this choice.

5. Computational cost
The paper lacks a quantitative evaluation of the additional computational cost introduced by the Sparsegen routing (such as the sorting operation and MLP prediction), and does not provide direct comparisons with baseline methods in terms of training/inference time, FLOPs, or latency.

References:
[1] Harder Task Needs More Experts: Dynamic Routing in MoE Models.
[2] HMoRA: Making LLMs More Effective with Hierarchical Mixture of LoRA Experts.
[3] KASA: Knowledge-Aware Singular-Value Adaptation of Large Language Models.
[4] MTL-LoRA: Low-Rank Adaptation for Multi-Task Learning.

**Questions:**

See the weaknesses.

---

> ### Author Response · Authors · 2025-11-22
>
> We thank the reviewer for the thoughtful suggestion. We will further refine the paper and re-upload a revised version based on the discussion and advice you provide.
>
> **W1. Insufficient Ablation Studies and Experiments**
> We conduct an ablation study of the different load balancing loss weights. So in general, the weight of the load balancing loss would affect the performance of the model. However, we do not observe significant systematic bias or instability when using different weights. According to the testing results, the different datasets exhibit slightly different preferences for the load balance loss coefficient, but the overall average accuracy changes only slightly from 83.52 to 84.56, which is within the absolute 1%.
>
> | $\alpha$ | ARC-C | ARC-E | Open | Comm | RTE | Avg. |
> | :--- | :---: | :---: | :---: | :---: | :---: | :---: |
> | 2.0 | 77.26 | 91.93 | 83.20 | 77.89 | 87.30 | 83.52 |
> | 1.0 | 78.67 | 92.11 | **85.00** | 79.30 | 87.72 | **84.56** |
> | 0.5 | 76.59 | 91.58 | 83.80 | **79.56** | **88.34** | 83.97 |
> | 0.1 | 77.93 | **92.28** | 83.40 | 78.92 | 86.95 | 84.15 |
> | 0.0 | **79.93** | 92.11 | 84.40 | 77.56 | 86.77 | 83.90 |
>
> *Table: Quantative result on different load balancing loss coeff $\alpha$ values (Qwen3-1.7B).*
>
> TopP:
> We used $p=0.4$ as recommended in the original paper and compare the performance of adapting the Top-P method in Mixture of LoRA Experts settings. Although it achieves the best performance on the commenseQA dataset, the average performance of total 9 datasets is slightly worse than the ReLU router, with total 0.7% lower than our proposed method. What is more, our proposed method does not have the requirements of the additional hyper-parameter p.
>
> | Method | Model | TP | MMLU_P | ARC-C | ARC-E | Open | Comm | SWAG | Hella | CoLA | RTE | Avg |
> | :--- | :--- | :---: | :---: | :---: | :---: | :---: | :---: | :---: | :---: | :---: | :---: | :---: |
> | MoLA(8888) | Qwen3-1.7B | 4.12 % | 51.12 | 76.59 | 88.60 | 82.40 | 76.49 | 84.11 | 83.35 | **83.89** | 86.64 | 79.24 |
> | MoLA(2468) | Qwen3-1.7B | 2.39 % | 49.96 | 76.92 | 88.42 | 83.00 | 75.84 | 84.17 | 87.09 | 83.60 | 84.48 | 79.28 |
> | ToP-P(p=0.4) | Qwen3-1.7B | 4.12 % | 52.08 | 78.00 | 91.43 | 84.80 | **79.71** | 86.24 | 87.30 | 82.63 | 86.00 | 80.91 |
> | ReMoLE | Qwen3-1.7B | 4.12 % | 53.40 | **79.60** | 91.75 | 84.80 | 79.44 | 86.37 | 88.00 | 82.12 | 83.74 | 81.02 |
> | LD-MoLE | Qwen3-1.7B | 4.23 % | **53.82** | 78.67 | **92.11** | **85.00** | 79.30 | **86.72** | **88.71** | 82.61 | **87.72** | **81.63** |
>
> *Table: Additional comparison between ToP-P method with $p=0.4$ across downstream tasks on Qwen3-1.7B.*
>
> HMoRA:
> For the HMoRA, we argue that the main focus of the HMoRA method is on the generalization ability. They integrating token-level and task-level routing in a hierarchical manner and introducing novel routing auxiliary loss. However, in our paper, we want to discuss the performance on downstream task with routing design itself, instead of increasing the input information and introducing new loss. The experiments setting of our paper is also different from HMoRA, unlike HMoRA finetuning on a subset of FlanV2 dataset and then evaluating on the different downstream tasks, our paper is focused on finetuning and evaluation on the same downstream task.

---

> ### Author Response · Authors · 2025-11-22
>
> **W2. Lack of Large-Scale Model Comparison**
> In order to further address the scalability of our method, we tested LD-MoLE on Llama3.1-8B with the best computational resources we could and compared it with selected baselines. For all downstream tasks, the LD-MoLE achieves the best performances on 7 out of 9 datasets. And for average scores, the LD-MoLE also makes 0.7% higher than ReMoLE.
>
> | Method | Model | TP | MMLU_P | ARC-C | ARC-E | Open | Comm | SWAG | Hella | CoLA | RTE | Avg |
> | :--- | :--- | :---: | :---: | :---: | :---: | :---: | :---: | :---: | :---: | :---: | :---: | :---: |
> | MoLA(8888) | Llama3.1-8B | 2.17 % | 48.21 | 76.79 | 86.32 | 86.40 | **84.11** | 89.15 | 94.06 | **87.73** | 89.53 | 82.48 |
> | MoLA(2468) | Llama3.1-8B | 1.26 % | 50.37 | 77.90 | 87.37 | 87.60 | 83.87 | 86.97 | 94.03 | 87.25 | 90.25 | 82.85 |
> | ReMoLE | Llama3.1-8B | 2.17 % | 54.98 | 82.00 | **91.61** | 87.60 | 83.17 | 92.22 | 94.95 | 85.27 | 88.43 | 84.47 |
> | LD-MoLE | Llama3.1-8B | 2.33 % | **55.98** | **83.67** | **91.61** | **88.00** | 83.01 | **92.29** | **95.45** | 85.25 | **91.32** | **85.18** |
> | MoLA(8888) | Llama3.2-3B | 3.11 % | 40.31 | 71.57 | 83.51 | 81.00 | 79.77 | 83.56 | 87.47 | 85.81 | **90.61** | 78.18 |
> | MoLA(2468) | Llama3.2-3B | 1.80 % | 42.31 | 71.91 | 83.86 | 83.60 | 80.02 | 83.96 | 87.31 | 86.00 | 89.53 | 78.72 |
> | ReMoLE | Llama3.2-3B | 3.11 % | 48.01 | **75.25** | 89.30 | 83.40 | 79.52 | 90.45 | 93.44 | 83.95 | 89.46 | 81.42 |
> | LD-MoLE | Llama3.2-3B | 3.28 % | **49.58** | 74.58 | **89.47** | **84.00** | **81.42** | **91.37** | **93.60** | **86.02** | 88.38 | **82.05** |
> | MoLA(8888) | Qwen3-1.7B | 4.12 % | 51.12 | 76.59 | 88.60 | 82.40 | 76.49 | 84.11 | 83.35 | **83.89** | 86.64 | 79.24 |
> | MoLA(2468) | Qwen3-1.7B | 2.39 % | 49.96 | 76.92 | 88.42 | 83.00 | 75.84 | 84.17 | 87.09 | 83.60 | 84.48 | 79.28 |
> | ReMoLE | Qwen3-1.7B | 4.12 % | 53.40 | **79.60** | 91.75 | 84.80 | **79.44** | 86.37 | 88.00 | 82.12 | 83.74 | 81.02 |
> | LD-MoLE | Qwen3-1.7B | 4.23 % | **53.82** | 78.67 | **92.11** | **85.00** | 79.30 | **86.72** | **88.71** | 82.61 | **87.72** | **81.63** |
>
> *Table: Comparison between methods across downstream tasks.*

---

> ### Author Response · Authors · 2025-11-22
>
> **W3. Unconventional Benchmark Selection**
> We appreciate the suggestion you provide and the dataset we evaluated on followed several well-known LoRA MoE paper:
> * HMora[1]: MMLU, MMLU-Pro, ARC-C, ARC-E, OpenBook, SWAG, CommonsenseQA
> * MoLA[2]: MRPC, COLA, RTE, ScienceQA, CommonsenseQA, OpenbookQA
> * AlphaLoRA[3]: MRPC, COLA, RTE, ScienceQA, CommonsenseQA, OpenbookQA
> * MixLoRA[4]: ARC-E, ARC-C, BoolQ, OpenBookQA, PIQA, SIQA, HellaSWAG, WinoGrande
>
> Our experiments are conducted in total 8 datasets including: ARC-C, ARC-E, CommonsenseQA, OpenbookQA, SWAG, HellaSWAG, CoLA, RTE which are highly overlapping with those tested in the above papers.
>
> For both KASA[5] and MTL-LoRA[6], both might share some similar intuitions with Mixture of LoRA experts settings, but our paper is mainly focused on the router design of Mixture of Experts(MoE) structure itself, which is not directly related to the two papers you mentioned. Moreover, we already included ARC-C, ARC-E, OpenbookQA, HellaSWAG, CoLA, RTE datasets which are also being used in two papers you mentioned. As your advices, we would include those two papers in our related works later.
>
> We also added MMLU_Pro as an additional dataset to test, and our proposed method achieves best performance from Qwen3-1.7B to Llama3.1-8B.
>
> [1] HMoRA: Making LLMs More Effective with Hierarchical Mixture of LoRA Experts
>
> [2] Higher Layers Need More LoRA Experts
>
> [3] AlphaLoRA: Assigning LoRA Experts Based on Layer Training Quality
>
> [4] MIXLORA: Enhancing Large Language Models Fine-Tuning with LoRA-based Mixture of Experts
>
> [5] KASA: Knowledge-Aware Singular-Value Adaptation of Large Language Models
>
> [6] MTL-LoRA: Low-Rank Adaptation for Multi-Task Learning

---

> ### Author Response · Authors · 2025-11-22
>
> **W4. Unclear Necessity of Sparsegen Routing**
> We value the constructive feedback and have carefully considered the concerns raised. Below we provide detailed responses to address the reviewer's concerns:
>
> **TopK and its variants**:
> TopK routing is inherently discontinuous: as the router scores vary smoothly during training, the selection remains fixed until two scores cross, at which point the routing decision switches abruptly. This creates a sharp discontinuity in the routing function, meaning the model output and therefore the gradient can change abruptly even for infinitesimal changes in the router scores. In practice, MoE implementations rely on a straight-through–style sparse gradient, which assumes a smooth dependence on the router scores and therefore cannot represent this abrupt behavior. Although TopK routing is widely used and generally stable, this abrupt switching remains an undesirable property because it misaligns the surrogate gradient with the true discrete routing behavior near the switch boundaries, which can lead to suboptimal or unstable routing decisions in practice.
> In contrast, continuous selection functions, e.g., Sparsegen, provide deterministic, continuous sparsity with well-defined gradients, which aligns the optimization dynamics with the routing behavior, leading to improved performance compared to TopK.
>
> **Gumbel-Softmax**:
> Gumbel-Softmax is not designed to encourage sparsity. It is fundamentally a stochastic categorical sampling approximation: it injects Gumbel noise and produces a soft probability vector on the simplex to approximate the non-differentiable argmax. In contrast, our routing formulation requires a deterministic and stable selection for downstream computation (expert activation and sparse gradient propagation). Using Gumbel-Softmax would produce stochastic, dense-like soft assignments, resulting in fluctuating expert choices and noisy gradients that can destabilize routing. Mixture-of-Experts models generally adopt deterministic subset selection for routing. This assumption is widely used in MoE architectures because it not only enhances the interpretability of the expert selection process, but also enables more efficient and scalable implementations in practice, as demonstrated in several prior works[4][5].
> Our Sparsegen-based method directly produces deterministic sparse simplex outputs—no sampling, no injected noise, and substantially more deterministic and significantly sparser outputs compared to the soft assignments produced by Gumbel-Softmax. This ensures that both the routing decisions and their gradients are natively sparse.
>
> **Soft routing**:
> For soft routing( directly using softmax without TopK selection ), it requires the activation of all experts. This approach might be suitable in the mixture of LoRA experts method since even activating all the lora Experts won’t add significant computational cost due to LoRA itself. In standard MoE models, using soft routing would result in substantial computational overhead. This has already been discussed in the HMoRA[1] Appendix A Routing Method.
>
> Although in this paper we demonstrate our routing design in Mixture of LoRA Experts settings, we still hope that we could extend our method in the pretraining settings in the future. That is also a reason why we choose sparsegen since it has properties including fully differentiable, sparse activation, directly fit with commonly used load balancing loss(DeepSeek[2], Qwen3[3]) and so on.
>
> [1] HMoRA: Making LLMs More Effective with Hierarchical Mixture of LoRA Experts
>
> [2] DeepSeekMoE: Towards Ultimate Expert Specialization in Mixture-of-Experts Language Models
>
> [3] Qwen3 Technical Report
>
> [4] Adaptive mixtures of local experts.
>
> [5] Hierarchical mixtures of experts and the em algorithm

---

> ### Author Response · Authors · 2025-11-22
>
> **ReLU-based differentiable routers**:
> For ReLU-based differentiable routers, it would have zero expert activation cases due to ReLU function. As you can see, if the gating scores are all less or equal to zero, then ReLU-based router would select no expert for that token at that layer. For the detailed discussion, please refer to our paper **Appendix C.1**. According to the ReMoE[1], the FFN is skipped for tokens without active experts, as the unpermutation operation for these tokens yields all zeros. In such cases, only the residual connection is applied. This typically occurs with common tokens such as ' ', `'\n'`, and 'the'. What's more, ReLU routing outputs can be arbitrarily small, making their own designed combination loss(the load balancing loss and regularization loss) term trivially bounded at 0. So the ReMoE[1] introduces an adaptive update parameter $\lambda_i$ to avoid routing collapse. But still, there would be two hyperparameters $\lambda_0$ and $\alpha$ for their own designed loss term.
>
> But in contrast, the sparsegen routing would guarantee that at least one expert being activated because of its closed-form solution. And also, with the sparsegen router, our proposed method could directly fit with the commonly used load balancing loss(DeepSeek[2], Qwen3[3]). In our paper table 1, we already compare the ReLU routing( ReMoLE ) as our baseline since this is the most related method considering its fully differentiability.
>
> We also provide a discussion in the Introduction of our paper to address the reason why we select the sparsegen function:
>
> We adopt Sparsegen[4] as the projection onto the probability simplex and propose a dynamic routing mechanism and the corresponding training pipeline that has the following benefits: (1) the closed-form formulation with Sparsegen to decide routing probability ensures differentiability and guarantees that every token is assigned to at least one expert; (2) the routing admits a well-defined subgradient; (3) the derivative of the routing is upper-bounded, facilitating stable optimization; and (4) the routing design supports sparse yet controllable allocations. Building on this foundation, we introduce a lightweight, shared multi-layer perceptron (MLP) that predicts a token-specific sparsity parameter $\lambda$, governing expert selection.
>
> [1] ReMoE: Fully Differentiable Mixture-of-Experts with ReLU Routing
>
> [2] DeepSeekMoE: Towards Ultimate Expert Specialization in Mixture-of-Experts Language Models
>
> [3] Qwen3 Technical Report
>
> [4] On Controllable Sparse Alternatives to Softmax

---

> ### Author Response · Authors · 2025-11-22
>
> **W5. Computational cost**
> We would like to clarify that our main contribution is the token-level and layer-level adaptive expert selection enabled by the proposed sparsegen activation function. Accordingly, our experiments primarily focus on reporting task accuracy and comparisons with state-of-the-art methods. To provide a more comprehensive evaluation of computational efficiency, we have additionally computed FLOPs, as shown in the table below.
>
> The FLOP analysis further demonstrates the effectiveness of the sparsity loss introduced in **Section 3.2.2**. As the sparsity-loss coefficient $\beta$ increases, the number of activated LoRA experts decreases, resulting in lower FLOPs. Nevertheless, compared with conventional TopK routing and ReLU routing, the main source of additional computation remains the shared MLP used to predict each token’s sparsity factor $\lambda$. To further reduce this cost, one potential direction is to replace the shared MLP with a lightweight gating linear projection to generate the sparsity coefficient $\lambda$. A discussion of how to reduce the cost is already included in our paper, **Appendix C.3**. Inspired by the discussion, as an additional supplemental design to the **Appendix C.3**, Increasing the output dimension of the gating score from N(number of experts) to $N+1$ and using the last element to directly pass a logsigmoid function might avoid the additional flop cost caused by learning $\lambda$. We will leave this to the future work for efficiency focus.
>
> | Qwen3-1.7B | MoLA(8888) | MoLA(2468) | ReMoLE | LD-MoLE($\beta=1.0$) | LD-MoLE($\beta=0.1$) | LD-MoLE($\beta=0.01$) | LD-MoLE($\beta=0.0$) |
> | :--- | :---: | :---: | :---: | :---: | :---: | :---: | :---: |
> | MFLOPs | 43 | 40 | 74 | 83 | 90 | 100 | 106 |
>
> *Table: Quantative result on the FLOP of different method. The FLOPs being calculated here are effective FLOPs including the router and LoRA experts. The backbone for the base LLMs is not counted since it would be the same for all different method.*

---

### Official Review · Reviewer_GWxU · 2025-10-30

**Soundness:** 3
**Presentation:** 3
**Contribution:** 2
**Rating:** 4
**Confidence:** 4

**Summary:**

This paper proposes LD-MoLE, a learnable dynamic routing framework for Mixture of LoRA Experts (MoLE) that replaces the traditional non-differentiable Top-K selection with a Sparsegen-based routing mechanism. The model adaptively determines the number of activated experts per token and per layer through a shared MLP predicting a sparsity factor λ, allowing token-dependent and layer-wise routing. The authors also introduce an analytical sparsity control loss to regularize the number of activated experts. Extensive experiments on Llama-3.2-3B and Qwen-3-1.7B show consistent improvements over MoLA (Top-K routing) and ReMoLE (ReLU-based routing) across multiple NLP benchmarks. The results demonstrate improved efficiency, stability, and interpretability of dynamic routing in parameter-efficient fine-tuning.

**Strengths:**

**Novel Differentiable Routing Mechanism:** The paper presents a clean and theoretically grounded alternative to Top-K routing using Sparsegen, providing both differentiability and guaranteed non-zero expert activation, addressing stability issues observed in prior dynamic routing work (e.g., ReMoE).

**Comprehensive Analysis:** The authors provide solid theoretical derivations, ablations on sparsity and λ prediction, and empirical validation on two base LLMs across 8 diverse benchmarks, clearly showing robustness and consistency.

**Strong Practical Value:** The design achieves dynamic, token-aware expert selection without increasing parameter count significantly (thanks to the shared MLP), making it practical for real-world PEFT and instruction-tuning applications.

**Weaknesses:**

1. **Gains are modest and transferability isn’t established.**
Reported improvements over Top-K/RELU routers (e.g., MoLA, ReMoLE) are incremental on small–mid LLMs and a limited task suite; the paper doesn’t convincingly show zero-shot / cross-dataset transfer (e.g., to harder or different domains such as math/code/MMLU) or robustness under distribution shift. Including a brief analysis (no extra training) of cross-dataset generalization would better support the method’s claims.

2. **No scale-up evidence (3B→8B→30B+) and limited backbone diversity.**
Experiments focus on small models (e.g., 1.7B–3B), leaving open whether learnable sparsity remains stable/efficient as parameters and depth grow (routing entropy, expert utilization balance, convergence speed, memory/latency). By contrast, scalable MoE work (e.g., Switch Transformer, DeepSeek-V3) validates routing and load balancing at tens–hundreds of billions of parameters; LoRA-MoE baselines like MixLoRA also report on a broader set of bases (e.g., LLaMA/Gemma/Mistral), aiding claims of generality. Adding at least partial scaling curves and results on a more diverse set of backbones would strengthen the paper.

3. **Why choose Sparsegen over other differentiable routers isn’t empirically settled.**
Theoretical appeal aside (deterministic, truly sparse, λ-controllable), the paper lacks a side-by-side comparison under the same budget against other differentiable routing families widely used in practice:
– Soft routing + Top-K variants from large-scale MoE (e.g., Switch/DeepSeek-V3 and modern aux-loss-free balancing);
– Gumbel-Softmax/Concrete reparameterization for differentiable categorical choices;
– ReLU-based differentiable routers.
A controlled ablation (same base/rank/FLOPs) on accuracy, stability, load balance, latency, and activated-expert count would clarify whether Sparsegen offers practical advantages beyond its theory.

**Questions:**

See above

---

> ### Author Response · Authors · 2025-11-22
>
> We thank the reviewer for the thoughtful suggestion. We will further refine the paper and re-upload a revised version based on the discussion and advice you provide.
>
> **W1. Gains are modest and transferability isn’t established.**
> Regarding the magnitude of performance gains, we emphasize that the improvement size is highly sensitive to experimental factors such as prompting style. In our early experiments, we observed that long or descriptive prompts introduced substantial variance in accuracy. To control for this factor, we standardized all evaluations using minimal prefix/suffix prompts, ensuring that the measured differences reflect model behavior rather than prompt artifacts.
>
> A second important source of variance arises from evaluation methodology. Several PEFT baselines compute accuracy by parsing generated text, a process that is sensitive to decoding randomness and formatting inconsistencies. To eliminate this confounding factor, we instead compute accuracy directly from the final-token logits over answer choices, which substantially reduces noise and leads to more stable and reliable comparisons.
>
> For transferability, we expanded our evaluation to include **MMLU-Pro**, providing a more challenging and cross-functional benchmark to assess our method. We note, however, that multi-task learning is not the primary focus of this work. Our central aim is to improve routing quality within MoE-based PEFT settings. That said, the proposed differentiable routing mechanism is inherently general and may support stronger cross-task generalization, particularly in scenarios where models must adapt expert usage across diverse task distributions. Exploring this broader multi-task or unseen cross-dataset applicability is a promising direction, though it lies beyond the scope of our current experiments.
>
> Overall, our method provides consistent gains in the PEFT setting, and the magnitude of improvement is comparable to those reported in recent Mixture-of-LoRA-Experts works [1, 2, 3, 4].
>
> [1] HMoRA: Making LLMs More Effective with Hierarchical Mixture of LoRA Experts
>
> [2] Higher Layers Need More LoRA Experts
>
> [3] AlphaLoRA: Assigning LoRA Experts Based on Layer Training Quality
>
> [4] MIXLORA: Enhancing Large Language Models Fine-Tuning with LoRA-based Mixture of Experts

---

> ### Author Response · Authors · 2025-11-22
>
> **W2. No scale-up evidence (3B→8B→30B+) and limited backbone diversity**
> Given the computational resources available to us, we focused our experiments on the Qwen3-1.7B and Llama3.2-3B models in the main paper, which are two model families also commonly used in prior works[1,2,3,4].
>
> We surveyed several Mixture of LoRA Experts papers and selected Qwen3 and Llama3 as representative model families that fit within our computational budget. Qwen3 is among the most advanced open-source LLMs, while Llama models are widely adopted as base architectures in Mixture of LoRA Experts settings[1,2,3,4].
> HMora[1]: Qwen2 1.5B.
> MoLA[2]: Llama2-7B.
> AlphaLoRA[3]: Llama-7B, Llama2-7B and Mistral-7B-v0.1.
> MixLoRA[4]: Gemma 2B, Llama2-7B, Llama3-8B and Llama2-13B.
>
> In order to further address the scalability of our method, we tested LD-MoLE on Llama3.1-8B with the best computational resources we could get and compared it with selected baselines. We also add MMLU_Pro as an additional datasets for your reference. We observe that for all downstream tasks, the LD-MoLE achieves the best performances on 7 out of 9 datasets. And for average scores, the LD-MoLE also makes $0.7\%$ higher than ReMoLE.
>
> | Model | Qwen3-1.7B | Llama3.2-3B | Llama3.1-8B |
> | :--- | :---: | :---: | :---: |
> | Improvement against MoLA(8888) | 2.39% | 3.87% | 2.7% |
> | Improvement against ReMoLE | 0.61% | 0.63% | 0.71% |
>
> *Table: Performance improvement versus MoLA(8888) and ReMoLE when scaling model size.*
>
> | Method | Model | TP | MMLU_P | ARC-C | ARC-E | Open | Comm | SWAG | Hella | CoLA | RTE | Avg |
> | :--- | :--- | :---: | :---: | :---: | :---: | :---: | :---: | :---: | :---: | :---: | :---: | :---: |
> | MoLA(8888) | Llama3.1-8B | 2.17 % | 48.21 | 76.79 | 86.32 | 86.40 | **84.11** | 89.15 | 94.06 | **87.73** | 89.53 | 82.48 |
> | MoLA(2468) | Llama3.1-8B | 1.26 % | 50.37 | 77.90 | 87.37 | 87.60 | 83.87 | 86.97 | 94.03 | 87.25 | 90.25 | 82.85 |
> | ReMoLE | Llama3.1-8B | 2.17 % | 54.98 | 82.00 | **91.61** | 87.60 | 83.17 | 92.22 | 94.95 | 85.27 | 88.43 | 84.47 |
> | LD-MoLE | Llama3.1-8B | 2.33 % | **55.98** | **83.67** | **91.61** | **88.00** | 83.01 | **92.29** | **95.45** | 85.25 | **91.32** | **85.18** |
> | MoLA(8888) | Llama3.2-3B | 3.11 % | 40.31 | 71.57 | 83.51 | 81.00 | 79.77 | 83.56 | 87.47 | 85.81 | **90.61** | 78.18 |
> | MoLA(2468) | Llama3.2-3B | 1.80 % | 42.31 | 71.91 | 83.86 | 83.60 | 80.02 | 83.96 | 87.31 | 86.00 | 89.53 | 78.72 |
> | ReMoLE | Llama3.2-3B | 3.11 % | 48.01 | **75.25** | 89.30 | 83.40 | 79.52 | 90.45 | 93.44 | 83.95 | 89.46 | 81.42 |
> | LD-MoLE | Llama3.2-3B | 3.28 % | **49.58** | 74.58 | **89.47** | **84.00** | **81.42** | **91.37** | **93.60** | **86.02** | 88.38 | **82.05** |
> | MoLA(8888) | Qwen3-1.7B | 4.12 % | 51.12 | 76.59 | 88.60 | 82.40 | 76.49 | 84.11 | 83.35 | **83.89** | 86.64 | 79.24 |
> | MoLA(2468) | Qwen3-1.7B | 2.39 % | 49.96 | 76.92 | 88.42 | 83.00 | 75.84 | 84.17 | 87.09 | 83.60 | 84.48 | 79.28 |
> | ReMoLE | Qwen3-1.7B | 4.12 % | 53.40 | **79.60** | 91.75 | 84.80 | **79.44** | 86.37 | 88.00 | 82.12 | 83.74 | 81.02 |
> | LD-MoLE | Qwen3-1.7B | 4.23 % | **53.82** | 78.67 | **92.11** | **85.00** | 79.30 | **86.72** | **88.71** | 82.61 | **87.72** | **81.63** |
>
> *Table: Comparison between methods across downstream tasks.*
>
> [1] HMoRA: Making LLMs More Effective with Hierarchical Mixture of LoRA Experts
>
> [2] Higher Layers Need More LoRA Experts
>
> [3] AlphaLoRA: Assigning LoRA Experts Based on Layer Training Quality
>
> [4] MIXLORA: Enhancing Large Language Models Fine-Tuning with LoRA-based Mixture of Experts

---

> ### Author Response · Authors · 2025-11-22
>
> **W3. Why choose Sparsegen over other differentiable routers isn’t empirically settled.**
> We value the constructive feedback and have carefully considered the concerns raised. Below we provide detailed responses to address the reviewer's concerns:
>
> **TopK and its variants**:
> TopK routing is inherently discontinuous: as the router scores vary smoothly during training, the selection remains fixed until two scores cross, at which point the routing decision switches abruptly.  This creates a sharp discontinuity in the routing function, meaning the model output and therefore the gradient can change abruptly even for infinitesimal changes in the router scores. In practice, MoE implementations rely on a straight-through–style sparse gradient, which assumes a smooth dependence on the router scores and therefore cannot represent this abrupt behavior. Although TopK routing is widely used and generally stable, this abrupt switching remains an undesirable property because it misaligns the surrogate gradient with the true discrete routing behavior near the switch boundaries, which can lead to suboptimal or unstable routing decisions in practice.
> In contrast, continuous selection functions, e.g., Sparsegen, provide deterministic, continuous sparsity with well-defined gradients, which aligns the optimization dynamics with the routing behavior, leading to improved performance compared to TopK.
>
> For the TopK variants including Switch Transformers[1], AdaMOE[3] and Ada-k routing[4], although they are proposing the different way of selecting the number of k, they would still face the same problem as TopK since the TopK part doesn't really change. And for aux-loss-free balancing[5], the paper still uses conventional TopK routing but with an expert-wise bias term added to the gating score.
>
> We have already discussed Switch Transformers[1] but we will include the following discussion in related works:
>
> The LongCat-Flash[1] still uses conventional TopK routing but with zero-computation experts simply returning the input as experts' output to avoid additional computational cost for simple tokens. What's more, LongCat-Flash[1] employs the Shortcut-connected MoE (ScMoE) architecture[4] to create a more substantial overlap window than shared-expert designs. AdaMOE[2] also uses conventional TopK routing with k larger than in vanilla MoE but achieves token-adaptive expert selection by incorporating null experts, which are defined as an empty operation. Ada-k routing[3] introduces an allocator and then obtains k* for customized expert resource allocation instead of fixed TopK through a non-differentiable sampling operation with a RL-based optimization framework. Loss-Free Balancing[5] adds an expert-wise bias term to the gating score to adjust the routing strategy by influencing the TopK selection.
>
> **Gumbel-Softmax**:
> Gumbel-Softmax is not designed to encourage sparsity. It is fundamentally a stochastic categorical sampling approximation: it injects Gumbel noise and produces a soft probability vector on the simplex to approximate the non-differentiable argmax. In contrast, our routing formulation requires a deterministic and stable selection for downstream computation (expert activation and sparse gradient propagation). Using Gumbel-Softmax would produce stochastic, dense-like soft assignments, resulting in fluctuating expert choices and noisy gradients that can destabilize routing. Mixture-of-Experts models generally adopt deterministic subset selection for routing. This assumption is widely used in MoE architectures because it not only enhances the interpretability of the expert selection process, but also enables more efficient and scalable implementations in practice, as demonstrated in several prior works[6][7].
> Our Sparsegen-based method directly produces deterministic sparse simplex outputs—no sampling, no injected noise, and substantially more deterministic and significantly sparser outputs compared to the soft assignments produced by Gumbel-Softmax. This ensures that both the routing decisions and their gradients are natively sparse.
>
> [1] Switch Transformers: Scaling to Trillion Parameter Models with Simple and Efficient Sparsity
>
> [2] Longcat-flash technical report
>
> [3] Adamoe: Token-adaptive routing with null experts for mixture-of-experts language models
>
> [4] Ada-k routing: Boosting the efficiency of moe-based llms
>
> [5] Auxiliary-Loss-Free Load Balancing Strategy for Mixture-of-Experts
>
> [6] Adaptive mixtures of local experts.
>
> [7] Hierarchical mixtures of experts and the em algorithm

---

> ### Author Response · Authors · 2025-11-22
>
> **Soft routing**:
> For soft routing( directly using softmax without TopK selection ), it requires the activation of all experts. This approach might be suitable in the mixture of LoRA experts method since even activating all the lora Experts won’t add significant computational cost due to LoRA itself. In standard MoE models, using soft routing would result in substantial computational overhead. This has already been discussed in the HMoRA[1] Appendix A Routing Method.
>
> Although in this paper we demonstrate our routing design in Mixture of LoRA Experts settings, we still hope that we could extend our method in the pretraining settings in the future. That is also a reason why we choose sparsegen since it has properties including fully differentiable, sparse activation, directly fit with commonly used load balancing loss(DeepSeek[2], Qwen3[3]) and so on.
>
> **ReLU-based differentiable routers**:
> For ReLU-based differentiable routers, it would have zero expert activation cases due to ReLU function. As you can see, if the gating scores are all less or equal to zero, then ReLU-based router would select no expert for that token at that layer. For the detailed discussion, please refer to our paper **Appendix C.1**. According to the ReMoE[4], the FFN is skipped for tokens without active experts, as the unpermutation operation for these tokens yields all zeros. In such cases, only the residual connection is applied. This typically occurs with common tokens such as ' ', `'\n'`, and 'the'. What's more, ReLU routing outputs can be arbitrarily small, making their own designed combination loss(the load balancing loss and regularization loss) term trivially bounded at 0. So the ReMoE[4] introduces an adaptive update parameter $\lambda_i$ to avoid routing collapse. But still, there would be two hyperparameters $\lambda_0$ and $\alpha$ for their own designed loss term.
>
> But in contrast, the sparsegen routing would guarantee that at least one expert being activated because of its closed-form solution. And also, with the sparsegen router, our proposed method could directly fit with the commonly used load balancing loss(DeepSeek[2], Qwen3[3]). In our paper table 1, we already compare the ReLU routing( ReMoLE ) as our baseline since this is the most related method considering its fully differentiability.
>
> **Further discussion**:
> We argue that only ReLU based routing is actually fully differentiable among all the "differentiable routing families" you mentioned in question 3, which we already adapted as ReMoLE in our table 1 as a baseline. We also provide an additional analysis of ReLU routing in our paper **Appendix C.1**.
>
> Achieving exactly the same budget across TopK, ReLU-routing, and LD-MoLE is non-trivial because both ReLU and our method perform token-level dynamic routing. Instead, we match base model, LoRA rank, number of experts, and training schedule, and we additionally report MFLOPs to show that all methods operate in a comparable regime.
>
> [1] HMoRA: Making LLMs More Effective with Hierarchical Mixture of LoRA Experts
>
> [2] DeepSeekMoE: Towards Ultimate Expert Specialization in Mixture-of-Experts Language Models
>
> [3] Qwen3 Technical Report
>
> [4] ReMoE: Fully Differentiable Mixture-of-Experts with ReLU Routing

---

> > ### Comment · Reviewer_GWxU · 2025-11-27
> >
> > I very much appreciate your evaluation setup – the design and choice of benchmarks make sense to me. However, I would like to better understand the ultimate goal behind improving routing quality in MoE-based PEFT settings.
> > From your current experiments, it seems the main benefit is improved single-task fine-tuning performance. Is that indeed the primary objective?
> > My confusion comes from how I typically think about MoE: its main advantage is to increase model capacity and, as a result, enhance generalization across tasks or distributions. In that sense, focusing on MoE purely for single-task PEFT feels somewhat unintuitive to me. Could you clarify whether your routing improvements are mainly intended for single-task performance gains, or whether you also envision broader benefits for multi-task or cross-task generalization (even if not fully explored in this paper)?

---

> > > ### Author Response · Authors · 2025-11-27
> > >
> > > Thank you for the thoughtful follow-up and we agree this is an important clarification.
> > >
> > > Our primary goal in this paper is to study and improve the *routing mechanism itself* by making it (i) fully differentiable and (ii) dynamically adaptive at the token/layer level. We use MoE-based PEFT as our experimental setting because it lets us isolate routing quality under a realistic MoE-style interface (expert selection + load balancing) without the prohibitive cost of pretraining a standard MoE model from scratch.
> > >
> > > **Is single-task PEFT the main objective?**
> > > In this paper, we intentionally focus on single-task PEFT fine-tuning as a controlled and practical setting to evaluate routing quality. Our goal is not to imply that MoE is only meaningful for single-task scenarios, but rather to isolate and demonstrate one important aspect of our method to improve the effectiveness of routing when only a small number of parameters are trainable.
> > > We chose single-task fine-tuning for the following reasons: It provides a clean, well-controlled environment for evaluating routing behavior. The base model, LoRA rank, number of experts, and training schedule can be held fixed, making performance differences more attributable to routing behavior rather than to multi-task data mixture or sampling ratios.
> > > Multi-task or cross-task training introduces confounding factors such as task imbalance, dataset mixing strategies, and optimization schedules. Take HMoRA[1] as an example, they fine-tuned their model on a subset of Flan v2 and then evaluated on several benchmarks. However, the proportion of training data, which is mentioned in the HMoRA paper, may have a significant impact on the training outcomes. Because these mixture details strongly influence results and can be difficult to reproduce exactly across implementations, we chose single-task fine-tuning to reduce variance and make the router comparison more reliable.
> > >
> > > **Do we envision broader benefits beyond single-task gains?**
> > > Yes, we fully agree that MoE routing is fundamentally motivated by enabling expert specialization, which naturally extends to multi-task and cross-task generalization. Our differentiable, dynamically sparse routing mechanism is designed with this broader goal in mind. Although comprehensive multi-task experiments are beyond the scope of this paper, we did include MMLU-Pro as an additional evaluation to probe cross-domain generalization on a challenging, mixed-distribution benchmark. The improvements we observe there suggest that our routing gains are not limited to a single-task PEFT setting. We view these results as an initial step toward understanding the broader applicability of our method, and we expect the benefits to extend to full multi-task or heterogeneous training scenarios.
> > >
> > > Finally, it is worth noting that even within our controlled setting, we observe that *fully differentiable dynamic routing* (ReLU-based routing[2] and our Sparsegen-based routing) consistently improves over conventional TopK routing on most benchmarks, suggesting that the routing mechanism itself is actually a key factor, independent of whether the training is single-task or multi-task.
> > >
> > > [1] HMoRA: Making LLMs More Effective with Hierarchical Mixture of LoRA Experts
> > >
> > > [2] ReMoE: Fully Differentiable Mixture-of-Experts with ReLU Routing

---

> > > > ### Comment · Reviewer_GWxU · 2025-11-28
> > > >
> > > > I understand your clarifications. Once the system is back to normal and allows score updates, I will increase my rating.

---

### Official Review · Reviewer_Ckcz · 2025-10-31

**Soundness:** 4
**Presentation:** 3
**Contribution:** 4
**Rating:** 8
**Confidence:** 3

**Summary:**

This paper proposes LD-MoLE, a novel **Learnable Dynamic** routing mechanism for **M**ixture **o**f **L**oRA **E**xperts. The method addresses key limitations in existing MoLE approaches by replacing the non-differentiable Top-K routing with a fully differentiable routing function based on Sparsegen projection. LD-MoLE introduces a lightweight shared MLP that predicts a token- and layer-dependent sparsity parameter (λ), enabling adaptive expert allocation. An analytical sparsity loss is derived to explicitly control the number of activated experts. Extensive experiments on Llama-3.2-3B and Qwen3-1.7B demonstrate state-of-the-art performance across diverse benchmarks, outperforming strong baselines like MoLA (Top-K) and ReMoLE (ReLU-based routing).

**Strengths:**

- The idea of combining Sparsegen-based differentiable routing with a learned λ-predicting MLP is novel and impactful. It elegantly solves the non-differentiability of Top-K and the instability of ReLU routing, advancing the MoLE paradigm.
- Theoretical grounding such as closed-form solutions, proofs is solid. Experiments are comprehensive, covering multiple models, tasks, and ablation studies including sparsity control, λ analysis and zero-activation issue.
- The paper is well-structured, clearly written, and figures effectively illustrate the methodology and findings.
- LD-MoLE provides a principled, efficient, and high-performing framework for dynamic expert routing, with potential influence on both PEFT and MoE research communities.

**Weaknesses:**

- **Computational Overhead**: While parameter-efficient, the computational cost (e.g., latency, FLOPs) of the dynamic routing mechanism—especially the Sparsegen projection—compared to simple Top-K is not quantified. A brief analysis would strengthen the efficiency claim.
- **Baseline Comparison**: Including a fixed-λ Sparsegen baseline would more directly isolate the contribution of the learned λ (via the MLP) from the contribution of Sparsegen itself.
- **Hyperparameter Sensitivity**: The impact of the new hyperparameters (α, β, and target sparsity k) on performance and stability is not discussed. A sensitivity analysis would improve practical usability.

**Questions:**

1. The shared MLP predicts λ for the same module type across all layers. Does this design potentially limit the model's ability to learn highly specialized, layer-specific routing strategies? What is the trade-off between parameter efficiency and layer-wise flexibility here?

2. In the sparsity control analysis (Sec. 4.4), using the sparsity loss (β>0) reduces expert count at a performance cost. Do you see a pathway for the model to learn a task-optimal sparsity level automatically, rather than relying on a pre-defined k?

---

> ### Author Response · Authors · 2025-11-22
>
> We thank the reviewer for the thoughtful suggestion. We will further refine the paper and re-upload a revised version based on the discussion and advice you provide.
>
> **W1. Computational Overhead.** We would like to clarify that our main contribution is the token-level and layer-level adaptive expert selection enabled by the proposed sparsegen activation function. Accordingly, our experiments primarily focus on reporting task accuracy and comparisons with state-of-the-art methods. To provide a more comprehensive evaluation of computational efficiency, we have additionally computed FLOPs, as shown in the table below.
>
> The FLOP analysis further demonstrates the effectiveness of the sparsity loss introduced in **Section 3.2.2**. As the sparsity-loss coefficient $\beta$ increases, the number of activated LoRA experts decreases, resulting in lower FLOPs. Nevertheless, compared with conventional TopK routing and ReLU routing, the main source of additional computation remains the shared MLP used to predict each token’s sparsity factor $\lambda$. To further reduce this cost, one potential direction is to replace the shared MLP with a lightweight gating linear projection to generate the sparsity coefficient $\lambda$. A discussion of how to reduce the cost is already included in our paper, **Appendix C.3**. Inspired by the discussion, as an additional supplemental design to the **Appendix C.3**, Increasing the output dimension of the gating score from N(number of experts) to $N+1$ and using the last element to directly pass a logsigmoid function might avoid the additional flop cost caused by learning $\lambda$.
>
> | Qwen3-1.7B | MoLA(8888) | MoLA(2468) | ReMoLE | LD-MoLE($\beta=1.0$) | LD-MoLE($\beta=0.1$) | LD-MoLE($\beta=0.01$) | LD-MoLE($\beta=0.0$) |
> | :--- | :---: | :---: | :---: | :---: | :---: | :---: | :---: |
> | MFLOPs | 43 | 40 | 74 | 83 | 90 | 100 | 106 |
>
> *Table: Quantative result on FLOPs of different method. The FLOPs being calculated here are effective FLOPs including the router and activated LoRA experts. The backbone for the base LLMs is not counted since it would be the same for all different method.*
>
> **W2. Baseline Comparison: Including a fixed-$\lambda$ Sparsegen baseline.** In our paper **section 4.3**, we have an ablation study to discuss fixed-$\lambda$ Sparsegen. Because the range of $\lambda$ is $(-\infty, 1)$, and for different datasets different $\lambda$ would have variance performance, we think it might be tricky to directly choose a fixed value of $\lambda$ to directly use it as a baseline. So we discuss fixed-$\lambda$ Sparsegen as an ablation study to show the advantage of learning $\lambda$.
>
> **W3. Hyperparameter Sensitivity** Thank you for the insightful and constructive suggestion! Here is the ablation study of the different load balancing loss weights. So in general, the weight of the load balancing loss would affect the performance of the model. However, we do not observe significant systematic bias or instability when using different weights. According to the testing results, the different datasets exhibit slightly different preferences for the load balance loss coefficient, but the overall average accuracy changes only slightly from 83.52 to 84.56, which is within the absolute 1%.
>
> | $\alpha$ | ARC-C | ARC-E | Open | Comm | RTE | Avg. |
> | :--- | :---: | :---: | :---: | :---: | :---: | :---: |
> | 2.0 | 77.26 | 91.93 | 83.20 | 77.89 | 87.30 | 83.52 |
> | 1.0 | 78.67 | 92.11 | **85.00** | 79.30 | 87.72 | **84.56** |
> | 0.5 | 76.59 | 91.58 | 83.80 | **79.56** | **88.34** | 83.97 |
> | 0.1 | 77.93 | **92.28** | 83.40 | 78.92 | 86.95 | 84.15 |
> | 0.0 | **79.93** | 92.11 | 84.40 | 77.56 | 86.77 | 83.90 |
>
> *Table: Quantative result on different load balancing loss coeff $\alpha$ values (Qwen3-1.7B).*

---

> ### Author Response · Authors · 2025-11-22
>
> Many thanks for your valuable questions, we put the responses in below:
>
> **Q1.**
> We put a discussion in **Appendix C.3** Additional Exploration on LD-MoLE. We think there would be potential trade off between learning layer-specific routing strategies, computational cost. As you can see from the detailed results in **Table 4** in the **Appendix C.3**, local MLP predicts $\lambda$ design would have a little lower performance compared to the shared structure. We doubt this is due to the fact that we only use one linear layer to predict $\lambda$ in local design. However, if you want to increase the number of layers to capture more complex relationships to learn highly specialized, layer-specific routing strategies would dramatically increase the total trainable parameter. This is actually the main reason we switch to the shared MLP design so we could balance the cost of trainable parameter and the ability of learning the highly specialized, layer-specific routing strategies.
>
> **Q2.**
> Thank you for the thoughtful question and this is indeed an interesting direction. One potential way might be to incorporate entropy-based regularization that encourages the router to learn its own sparsity pattern by directly minimizing the entropy of the routing distribution. Such objectives may allow the model to adaptively choose the number of experts per task without explicitly fixing k.
>
> However, these approaches generally provide no guarantee on the actual number of activated experts and may still result in high expert usage after training, especially for complex tasks. In contrast, our method offers explicit control over the sparsity level and does so without significant performance degradation. Overall, we view adaptive sparsity learning as a promising complementary direction, but our approach provides a practical and reliable trade-off in the current setting.

---

### Official Review · Reviewer_5J1o · 2025-11-01

**Soundness:** 3
**Presentation:** 3
**Contribution:** 3
**Rating:** 6
**Confidence:** 3

**Summary:**

The paper proposes LD-MoLE, a learnable dynamic routing mechanism for Mixture of LoRA Experts. Built on Sparsegen, the method introduces a differentiable and token-wise adaptive expert allocation strategy, with a learnable sparsity parameter predicted by a shared MLP. The framework also includes a sparsity control objective to regulate the number of activated experts. Experiments across several downstream tasks show improved performance compared to TopK and ReLU-based routing baselines.

**Strengths:**

The paper presents a clear motivation for improving expert allocation in MoE models and proposes a differentiable routing mechanism with controllable sparsity. The method is well-integrated into the LoRA-based MoE setting, and the overall presentation is clear. Experiments cover multiple tasks and include comparisons against relevant baselines, along with some ablation studies.

**Weaknesses:**

* **Applicability to Standard MoE Architectures**: While LD-MoLE is instantiated within the MoLE setting, it remains unclear whether the proposed routing mechanism can be seamlessly integrated into conventional MoE architectures with full FFN experts or applied during pretraining. Clarifying its plug-and-play compatibility with standard MoE training and inference pipelines would help better demonstrate the generality and practical advantages of the approach.

* **Insufficient Efficiency Evidence**: The paper claims that LD-MoLE effectively reduces the number of activated experts, but the evaluation largely reports task accuracy and qualitative activation plots. To substantiate efficiency, more concrete system metrics (e.g., FLOPs, latency, throughput and peak memory) would better support the conclusion that reduced activation translates into real compute savings.

* **Missing Discussion Regarding Sparse Routing.**: the paper lacks comparison with prior methods that target similar goals through dynamic routing or explicit sparsity control [1-3]. A clearer conceptual and empirical comparison with related works would strengthen the contribution.

[1] Team M L C, Li B, Lei B, et al. Longcat-flash technical report[J]. arXiv preprint arXiv:2509.01322, 2025.

[2] Zeng Z, Miao Y, Gao H, et al. Adamoe: Token-adaptive routing with null experts for mixture-of-experts language models[J]. arXiv
preprint arXiv:2406.13233, 2024.

[3] Yue T, Guo L, Cheng J, et al. Ada-k routing: Boosting the efficiency of moe-based llms[C]//The Thirteenth International Conference on Learning Representations. 2024.

**Questions:**

Please refer to Weaknesses

---

> ### Author Response · Authors · 2025-11-22
>
> We thank the reviewer for the thoughtful suggestion. We will further refine the paper and re-upload a revised version based on the discussion and advice you provide.
>
> **W1. Applicability to Standard MoE Architectures.**
> Yes, our method could easily integrate with conventional MoE architectures with full FFN experts and be applied during pretraining. As shown in Figure 1, the main structure we proposed here is a novel differentiable adaptive routing function, which is designed to plug naturally into existing MoE pipelines by replacing the LoRA module into FFN experts.
>
> The main reason we present our method in the current setting is that full-scale MoE training requires substantial computational sources that beyond our reach. We leave the extension of our approach to MoE pretraining as an important and promising direction for future work, and we will include a discussion of this in the revised manuscript.
>
> **W2. Insufficient Efficiency Evidence.**
> We would like to clarify that our main contribution is the token-level and layer-level adaptive expert selection enabled by the proposed sparsegen activation function. Accordingly, our experiments primarily focus on reporting task accuracy and comparisons with state-of-the-art methods. To provide a more comprehensive evaluation of computational efficiency, we have additionally computed FLOPs, as shown in the table below.
>
> The FLOP analysis further demonstrates the effectiveness of the sparsity loss introduced in **Section 3.2.2**. As the sparsity-loss coefficient $\beta$ increases, the number of activated LoRA experts decreases, resulting in lower FLOPs. Nevertheless, compared to conventional TopK routing and ReLU routing, the main source of additional computation remains the shared MLP used to predict each token’s sparsity factor $\lambda$. To further reduce this cost, one potential direction is to replace the shared MLP with a lightweight gating linear projection to generate the sparsity coefficient $\lambda$. A discussion of how to reduce the cost is included in our paper, **Appendix C.3**. Inspired by the discussion, as an additional supplemental design to the **Appendix C.3**, increasing the output dimension of the gating score from N(number of experts) to $N+1$ and using the last element to directly pass a logsigmoid function might avoid the additional flop cost caused by learning $\lambda$.
>
> | Qwen3-1.7B | MoLA(8888) | MoLA(2468) | ReMoLE | LD-MoLE($\beta=1.0$) | LD-MoLE($\beta=0.1$) | LD-MoLE($\beta=0.01$) | LD-MoLE($\beta=0.0$) |
> | :--- | :---: | :---: | :---: | :---: | :---: | :---: | :---: |
> | MFLOPs | 43 | 40 | 74 | 83 | 90 | 100 | 106 |
>
> *Table: Quantative result on the FLOP of different method. The FLOPs being calculated here are effective FLOPs including the router and LoRA experts. The backbone for the base LLMs is not counted since it would be the same for all different method.*

---

> ### Author Response · Authors · 2025-11-22
>
> **W3. Missing Discussion Regarding Sparse Routing**
> We thank the reviewer for directing us to these additional works. We will integrate a discussion of them into the related work section. After carefully reviewing the three papers, we note that each of them uses TopK routing or a modified form of TopK routing:
>
> The LongCat-Flash[1] still uses conventional TopK routing but with zero-computation experts simply returning the input as experts' output to avoid additional computational cost for simple tokens. What's more, LongCat-Flash[1] employs the Shortcut-connected MoE (ScMoE) architecture[4] to create a more substantial overlap window than shared-expert designs. AdaMOE[2] also uses conventional TopK routing with k larger than in vanilla MoE but achieves token-adaptive expert selection by incorporating null experts, which are defined as an empty operation. Ada-k routing[3] introduces an allocator and then obtains $k^*$ for customized expert resource allocation instead of fixed TopK through a non-differentiable sampling operation with a RL-based optimization framework.
>
> Discussions of TopK's discontinuity:
>
> TopK routing is inherently discontinuous: as the router scores vary smoothly during training, the selection remains fixed until two scores cross, at which point the routing decision switches abruptly. This creates a sharp discontinuity in the routing function, meaning the model output and hence the gradient can change abruptly even for infinitesimal changes in the router scores. In practice, MoE implementations rely on a straight-through–style sparse gradient, which assumes a smooth dependence on the router scores and therefore cannot represent this abrupt behavior. Although TopK routing is widely used, this abrupt switching remains an undesirable property because it misaligns the surrogate gradient with the true discrete routing behavior near the switch boundaries, which can lead to suboptimal or unstable routing decisions in practice.
> In contrast, continuous routing functions, e.g., Sparsegen, provide deterministic, continuous sparsity with well-defined gradients, which aligns the optimization dynamics with the routing behavior, leading to improved performance over TopK in our experiments.
>
> [1] Longcat-flash technical report
>
> [2] Adamoe: Token-adaptive routing with null experts for mixture-of-experts language models
>
> [3] Ada-k routing: Boosting the efficiency of moe-based llms
>
> [4] Shortcut-connected Expert Parallelism for Accelerating Mixture of Experts

---

### Author Response · Authors · 2025-11-27

Dear Reviewers,

Thank you for your constructive feedback and for the opportunity to strengthen our work. In response to your comments, we have uploaded a revised version and provided detailed point-by-point answers in the rebuttal. We would greatly appreciate it if you could take a look.

If you have any additional suggestions or questions, we would be happy to discuss them and further refine the paper.

Thank you again for your time and feedback.

---

### Meta-Review · Area_Chair_6Fdt · 2026-01-07

**Summary:**

This paper proposes LD-MoLE, a learnable dynamic routing mechanism for Mixture of LoRA Experts that replaces non-differentiable TopK routing with a Sparsegen-based differentiable routing function. The method introduces a shared MLP to predict token-specific and layer-wise sparsity parameters (λ), enabling adaptive expert allocation. An analytical sparsity control objective regularizes activated experts. Experiments on Qwen3-1.7B, Llama-3.2-3B, and Llama-3.1-8B demonstrate consistent improvements over TopK and ReLU-based routing baselines.

The paper makes a solid contribution to MoE routing methodology with a principled, differentiable approach that addresses known limitations of TopK and ReLU routing. The theoretical foundation is sound, experiments are now comprehensive (including 8B model and MMLU-Pro), and the rebuttal effectively addressed reviewer concerns. While gains are incremental, this reflects the mature state of the field, and consistent improvements across multiple models and benchmarks demonstrate the method's reliability.

**Reviewer Concerns:**

Most of concern has been addressed

**Reviewer Scores:**

Most of concern has been addressed. Reviewer would increase the scores

---

### Decision · Program_Chairs · 2026-01-26

Accept (Poster)